# Multivariate characterisation of morpho-biometric traits of indigenous helmeted Guinea fowl (*Numida meleagris*) in Nigeria

Abdulmojeed Yakubu[1]*, Praise Jegede[1,2], Mathew Wheto[3], Ayoola J. Shoyombo[4]*,
Ayotunde O. Adebambo[3]*, Mustapha A. Popoola[5], Osamede H. Osaiyuwu[6], Olurotimi
A. Olafadehan[7], Olayinka O. Alabi[4], Comfort I. Ukim[5], Samuel T. Vincent[1], Harirat
L. Mundi[1,8], Adeniyi Olayanju[4], Olufunmilayo A. Adebambo[3]

1 Department of Animal Science/Centre for Sustainable Agriculture and Rural Development, Faculty of
Agriculture, Nasarawa State University, Keffi, Shabu-Lafia Campus, Lafia, Nigeria, 2 National Biotechnology
Development Agency, Abuja, Nigeria, 3 Department of Animal Breeding and Genetics, Federal University of
Agriculture, Abeokuta, Nigeria, 4 Department of Animal Science, Landmark University, Omu-Aran, Nigeria,
5 Tertiary Education Trust Fund, Abuja, Nigeria, 6 Department of Animal Science, Faculty of Agriculture,
University of Ibadan, Ibadan, Nigeria, 7 Department of Animal Science, Faculty of Agriculture, University of
Abuja, Abuja, Nigeria, 8 Department of Animal Science, Faculty of Agriculture, Federal University of Lafia,
Lafia, Nigeria

* abdulkubu@nsuk.edu.ng (AY); shoyombo.ayoola@lmu.edu.ng (AJS); tumininuadebambo@gmail.com
(AOA)

pone.0261048

Universitesi, TURKEY

**Data Availability Statement:** All relevant data are
within the paper and its Supporting Information
files.

## Abstract

This study was conducted to characterise phenotypically helmeted Guinea fowls in three
agro-ecologies in Nigeria using multivariate approach. Eighteen biometric characters, four
morphological indices and eleven qualitative physical traits were investigated in a total of
569 adult birds (158 males and 411 females). Descriptive statistics, non-parametric Krus-
kal–Wallis H test followed by the Mann–Whitney U and Dunn-Bonferroni tests for post hoc,
Multiple Correspondence Analysis (MCA), Univariate Analysis, Canonical Discriminant
Analysis, Categorical Principal Component Analysis and Decision Trees were employed to
discern the effects of agro-ecological zone and sex on the morphostructural parameters.
Agro-ecology had significant effect (P<0.05; P<0.01) on all the colour traits. In general, the
most frequently observed colour phenotype of Guinea fowl had pearl plumage colour
(54.0%), pale red skin colour (94.2%), black shank colour (68.7%), brown eye colour
(49.7%), white earlobe colour (54.8%) and brown helmet colour (72.6%). The frequencies of
helmet shape and wattle size were significantly influenced (P<0.01) by agro-ecology and
sex. Overall, birds from the Southern Guinea Savanna zone had significantly higher values
(P<0.05) for most biometric traits compared to their Sudano-Sahelian and Tropical Rainfor-
est counterparts. They were also more compact (120.00 vs. 110.00 vs. 107.69) but had
lesser condition index (7.66 vs. 9.45 vs. 9.30) and lower long-leggedness (19.71 vs. 19.23
vs. 9.51) than their counterparts from the two other zones. Sexual dimorphism (P<0.05) was
in favour of male birds especially those in Southern Guinea Savanna and Sudano-Sahelian
zones. However, the MCA and discriminant analysis revealed considerable intermingling of
the qualitative physical traits, biometric traits and body indices especially between the
Sudano-Sahelian and Tropical Rainforest birds. In spite of the high level of genetic

**Funding:** The following authors: AY, MW, AJS, AOA, MAP, OHO, OAO, OOA, CIU,AO, OAA received funding through grant no TEF/DR&D/CE/NRF/UNI/ABEOKUTA/ STI/VOL.1. from the Tertiary Education Trust Fund (TETFUND) of the Federal Republic of Nigeria (https://tetfundserver.com/). The funders had no role in study design, data collection and analysis, decision to publish, or preparation of the manuscript.

**Competing interests:** The authors have declared that no competing interests exist.

admixture, the Guinea fowl populations could to a relative extent be distinguished using wing length, body length and eye colour. Generally, the birds from the three zones appeared to be more homogeneous than heterogeneous in nature. However, further complementary work on genomics will guide future selection and breeding programs geared towards improving the productivity, survival and environmental adaptation of indigenous helmeted Guinea fowls in the tropics.

## Introduction

Poultry species serve as important sources of animal protein and household income, especially for low-input and marginalized rural communities [1]. The helmeted Guinea fowl (*Numida meleagris*) belongs to the Galliformes order and the Numididae family. The game bird is terrestrial and commonly found in Africa [2]. The birds are indigenous to West Africa North of the Equatorial forest and are believed to have originated from the coast of Guinea in West Africa [3]. Based on evidence from archaeozoology and art, it was suggested that Mali and Sudan were centres of domestication of this species which might have occurred about 2,000 years BP [4]. In Nigeria, the Guinea fowl is a common game bird found mainly in the savanna region of northern Nigeria [5]. Guinea fowl farmers are basically involved in three major production systems: These include the Extensive System (Free range), Semi-intensive System (Partial confinement) and the Intensive System (Complete enclosure) [6]. In comparison with chicken, guinea fowl is economically more attractive in the tropics because it is not very demanding in terms of its diet, more rustic and adapts better to traditional farming system [7–9]. Guinea fowl is also highly valued for its meat and eggs. The meat is rich in vitamins and contains less cholesterol and fats, thereby making it a high quality protein source [10]. Additionally, the bird is used for different cultural purposes, and plays a role in poverty reduction among rural dwellers [11]. The bird also breeds seasonally and reaches its peak breeding activity during the summer period [12].

Every livestock species or breed is a real component of the animal genetic diversity of the world that deserves immense attention [13]. Despite the usefulness of Guinea fowl, it is poorly characterised in the tropics. This has limited its value as an unexploited potential for economic and industrial growth. Therefore, there is a need for proper characterisation geared mainly towards improvement in meat and egg production. The first step in such characterisation as outlined by FAO [14] involves the use of phenotypic characteristics which are aspects of physical appearance or other body parameters that can be measured qualitatively, and quantitatively. Variations in phenotypes have remained [15], and tolerance or susceptibility of birds to stressful environment could be linked to their phenotypic traits [16, 17]; hence, the need to understand such phenotypic diversity in the helmeted Guinea fowls especially in populations that have adapted to local environmental conditions. Under resource-poor settings, phenotypic approach is fundamental in livestock management because it is simple, fast, and cost-effective [18]. Also, morpho-biometrical characterisation (qualitative and quantitative traits) enables proper selection of elite animals, breeding, conservation and sustainable use of indigenous animal resources [19, 20]. Qualitative physical traits such as plumage colour, skin colour, shank colour, eye colour, helmet shape, wattle possession and skeleton structure are useful to farmers and breeders for identification and classification of Guinea fowl and to meet consumer preferences for specific phenotypic traits [21]. On the other hand, biometric measurements such as body weight, body length, chest circumference, wing length, wingspan and shank length are useful in

breeding programs, to revaluate local breeds, allow the preservation of animal biodiversity and support consumer demands [22, 23]. When such morphometric traits are considered jointly, multifactorial analyses have been shown to assess better the within-population variation which can be utilized in the discrimination of different population types [22, 24].

In Nigeria, south Saharan Africa, there is dearth of information on the phenotypic diversity of Guinea fowls [25]. The current study aimed to find differences in indigenous Guinea fowl based on qualitative physical traits, biometric traits and morphological indices in three agro-ecological zones in Nigeria. The knowledge of the morpho-biometrical traits will support the implementation of breeding and conservation strategies in order to guarantee the survival and continuous production of the Guinea fowl genetic resource in the tropics for improved food security and livelihoods.

## Materials and methods

### Ethics statement

In order to properly carry out the research, we adhered strictly to the ethical guidelines of the global code of conduct for research in resource-poor settings [26], following the Convention on Biological Diversity and Declaration of Helsinki. Although the study did not involve collection of blood and other tissue samples, we obtained field approval from the Research and Publication Directorate of Nasarawa State University, Keffi through permit no NSUK/FAC/ANS/GF100. Written informed consent was also obtained from each participating farmer in line with best global practices.

### Study area

The study was carried out in three agro-ecological zones of Nigeria namely; Sudano-Sahelian zone (Bauchi and Kano States), Southern Guinea Savanna zone (Nasarawa State and Abuja) and Tropical Rainforest zone (Ogun and Oyo States) (Fig 1). The Sudano-Sahelian zone is located between latitudes 10˚N and 14˚N and longitudes 4˚E and 14˚E, and lies immediately to the south of Sahara desert. The rainfall in this zone is less than 1000 mm per annum [27]. Temperature is high throughout the year with a mean minimum value of about 23˚C and mean maximum of about 34˚C. The zone is characterized by semi-arid grasslands vegetation while the density of trees and other plants decrease as one moves northwards. The Southern Guinea Savanna (GS) is part of the wider GS zone found on latitudes 7˚ and 10˚N and longitudes 3˚ and 14˚E [28]. The average annual maximum temperature ranges from 31 to 35˚C while the average annual minimum temperature is between 20 and 23˚C. It has mean annual rainfall of at least 1,600 millimeters and lowest mean monthly relative humidity of not less than 70 percent. It is a belt of mixture of trees and tall grasses. The Tropical Rainforest zone lies between latitudes 5.91˚ and 9.29˚N and longitude 2.79˚ and 6.11˚E [29]. Temperature ranges between 21˚C and 34˚C while the annual rainfall ranges between 1500 mm and 3000 mm. The vegetation consists of fresh water swamp and mangrove forest at the belt, the lowland forest, and secondary forest.

### Sampling procedure

A total of 569 adult (8 months old) Nigerian indigenous Guinea fowls comprising 109 birds (27 males and 82 females) from Southern Guinea Savanna zone, 270 birds (80 males and 190 females) from Sudano-Sahelian zone and 190 birds (51 males and 139 females)from Tropical Rainforest zone were used in the study. The indigenous birds were randomly sampled in smallholder rural farmers flocks and managed under the traditional low-input settings.

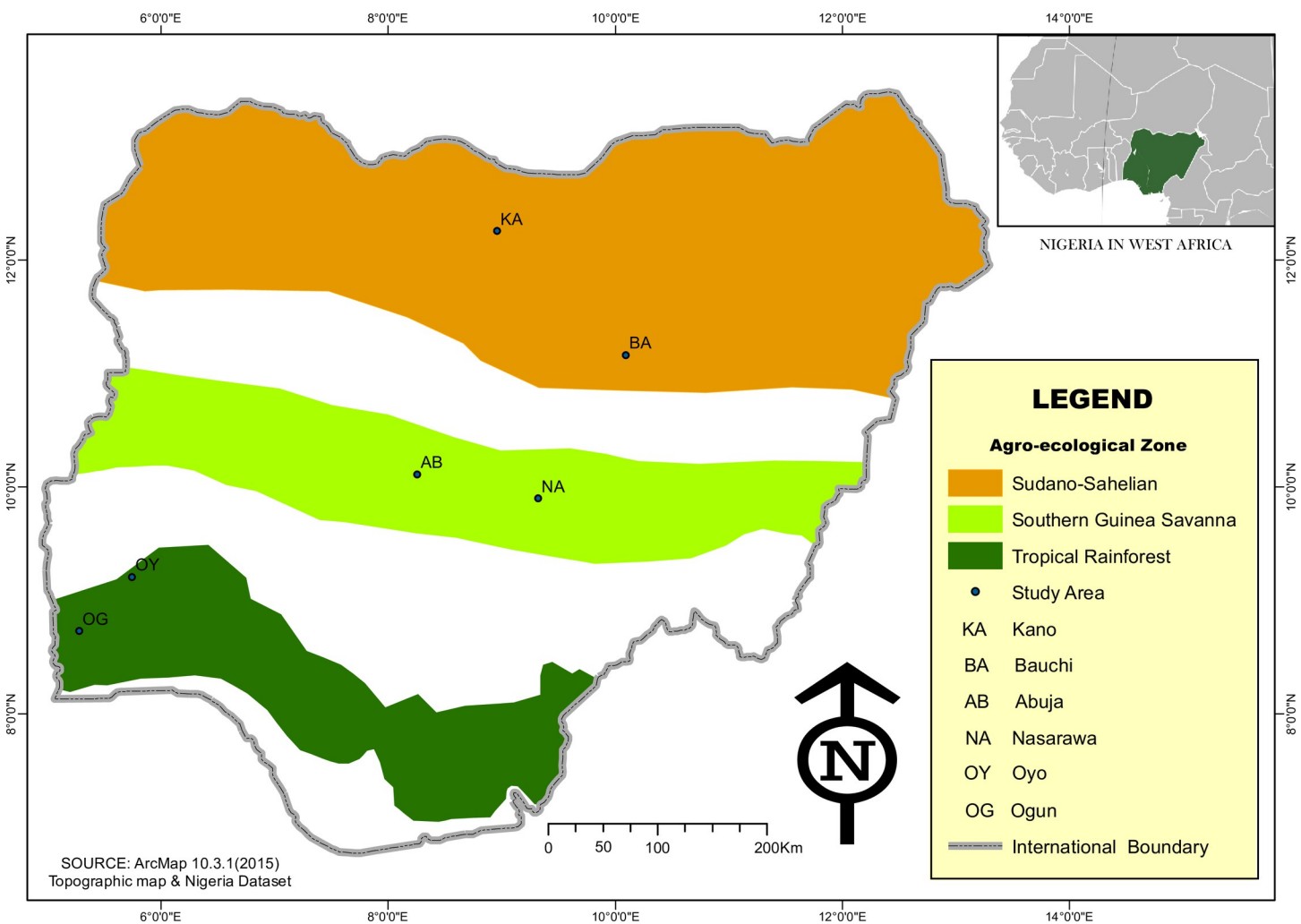

**Fig 1. Map of the three agro-ecological zones of study in Nigeria.**

Multistage sampling procedure was purposively and randomly adopted in the selection of States, Local Government Areas (LGAs), villages and Guinea fowl keepers in each agro-ecological zone. States, LGAs, and villages were purposively selected based on the knowledge of the availability of Guinea fowls in the communities as provided by the local Extension Agents and Community Heads. The number of sampling locations varied with 4 LGAs and 11 villages in the Southern Guinea Savanna, 5 LGAs and 15 villages in the Sudano-Sahelian, and 4 LGAs and 13 villages in the Tropical Rainforest. Based on willingness to participate in the research, eleven individuals were then randomly selected from each village making a total of 429 households (n = 121, 165 and 143 for Southern Guinea Savanna, Sudano-Sahelian and Tropical Rainforest, respectively).

## Data collection

Data collection was done in the rainy season (April to June, 2020). Morphologically distinct Guinea fowls were identified using phenotypic traits based on the standard descriptors by FAO [14], AU-IBAR [30] and the colour chart of Guinea fowl by GFIA [31]. The sexes were distinguished through visualisation of the vent and the use of helmet shape as well as wattle size and

shape [25]. Eleven qualitative physical parameters such as plumage colour, skin colour, shank colour, eye colour, earlobe colour, helmet colour, helmet shape, wattle possession, wattle size, wattle shape and skeletal structure were used to characterize the Guinea fowls morphologically. For quantitative (biometric) description, the following body parts were measured:

Body weight (kg): The live weight of the Guinea fowl;

Head length (cm): Taken between the most protruding point of the occipital and the frontal (lacrimal) bone;

Head thickness (cm): Head thickness measured as the circumference at the middle of the head;

Helmet length (cm): Measured as the distance between the base of the head to the tip of the helmet;

Helmet width (cm): Measured as the distance between the broadest part of the helmet;

Wattle length (cm): Taken as the distance between the base of the beak and the tip of the wattle;

Wattle width (cm): Measured as the distance between the broadest part of the wattle;

Neck length (cm): Distance between the occipital condyle and the cephalic borders of the coracoids;

Neck circumference (cm): Taken at the widest point of the neck;

Wing length (cm): Taken from the shoulder joint to the extremity of the terminal phalanx, digit III;

Wing span (cm): Distance between the two wings when stretched out;

Body length (cm): The distance from the first cervical vertebra (atlas) to the posterior end of the ischium;

Trunk length (cm): The distance between shoulder joint and posterior edge of the ischium;

Keel length (cm): Keel length (sternum or breast bone) measured from the anterior point of the keel to the posterior end;

Chest circumference (cm): Taken as the circumference of the body around the breast region;

Thigh length (cm): Distance between the hock joint and the pelvic joint;

Shank length (cm): Measured as the distance between the foot pad and the hock joint; and

Shank thickness (cm): Measured as the circumference at the middle or widest part of the shank.

Also, the following morphological indices were estimated [32]:

Massiveness: The ratio of live body weight to trunk length x 100;

Compactness: The ratio of chest circumference to trunk length x 100;

Long-leggedness: The ratio of shank length to body length x 100; and

Condition index: The ratio of live body weight to wing length $\times$ 100.

The weight measurement was taken using a hanging digital scale (WeiHeng Brand), the width measurements were taken using a vernier caliper (0.01 mm precision) while the length and circumference measurements were taken using a flexible tape measure.

## Statistical analysis

**Descriptive statistics.** Descriptive statistics were computed to determine the frequencies of the qualitative physical traits. Where statistical significant differences in the frequencies were obtained at agro-ecological and sex levels, they were assessed using the non-parametric Kruskal–Wallis H test followed by the Mann–Whitney U test for post hoc separation [33] of IBM-SPSS software [34]. This approach was adopted as a result of the small and unequal sample sizes of phenotypic groups including non-normality of the data distribution.

**Correspondence analysis.** Multiple correspondence analysis (MCA) was used to establish the relationships between the qualitative physical traits using JMP 16 [35] statistical software. In order to run the MCA, the input data (qualitative physical traits including their classes) were saved in IBM-SPSS software and opened under the JMP input file platform. Then, MCA was selected under multivariate methods. Preliminary analysis revealed that wattle possession and skeleton structure had zero variance and were excluded from the MCA.

**Univariate analysis.** Biometric traits and morphological indices were tested for normality with the Shapiro-Wilk's test (P<0.05) and by visual inspection of the histograms. Levene's test was used to confirm homogeneity of variances (P>0.05) as decribed by Brown et al. [36]. Due to small and unequal sizes, low male-female ratio and non-normality of the distribution of the data, the non-parametric Kruskal-Wallis H test was performed to compare mean ranks of biometric traits and morphological indices based on agro-ecology, sex, and sexes within each agro-ecology. In the case of significant Kruskal-Wallis H test, Dunn-Bonferroni test (agro-ecology) and Mann–Whitney U test (sex, and sexes within each agro-ecology) were used for pairwise comparisons of mean ranks.

**Stepwise canonical discriminant analysis.** Canonical discriminant analysis [37] option of IBM-SPSS [34] statistical software was applied to classify birds in the three agro-ecological zones based on quantitative traits. In the analysis, all the eighteen biometric traits and four morphological indices (covariates) were entered in a stepwise fashion as explanatory variables to establish and outline population clusters [38] based on agro-ecology. F-to-remove statistics was the criterion for variables' selection while multicollinearity was detected among the variables in the discriminant function using tolerance statistics. The ability of this discriminant model to identify birds in the Southern Guinea Savanna, Sudano-Sahelian and Tropical Rainforest zones was indicated as the percentage of individuals correctly classified from the sample that generated the model. The accuracy of the classification was evaluated using split-sample validation (cross-validation).

**Categorical principal component analysis.** Categorical principal component analysis (CATPCA) procedure was employed to explore hidden relationships among the qualitative physical traits (with the exception of wattle possession and skeleton structure due to zero variance), biometric traits and morphological as described by Martin-Collado et al. [39]. This was to allow for appropriate grouping of the guinea fowls based on agro-ecology and sex. The PCs were extracted based on Eigenvalues greater than 1 criterion. The convergence was 0.00001 with maximum iterations of 100. The PC matrix was rotated using the varimax criterion with Kaiser Normalization to facilitate easy interpretation of the analysis. The reliability of the PCA was tested using Chronbach's alpha using IBM-SPSS [34].

**Decision trees.** Chi-square automatic interaction detection (CHAID) and Exhaustive CHAID algorithms were employed to assign the birds into agro-ecological zones using the qualitative physical traits (with the exception of wattle possession and skeleton structure due to zero variance), biometric traits and morphological indices as the predictor variables. CHAID is a tree-based model with merging, partitioning and stopping stages that recursively uses multi-way splitting procedures to form homogenous subsets using Bonferroni adjustment

until the least differences between the predicted and actual values in a response variable are obtained [40]. It produces terminal nodes and finds the best possible variable or factor to split the node into two child nodes. The Exhaustive CHAID, as a modification of CHAID algorithm, applies a more detailed merging and testing of predictor variables [41]. The accuracy of CHAID and Exhaustive CHAID models was obtained from the percentage of individuals correctly classified in each agro-ecological zone. The predictive performance of each model was assessed using the goodness-of-fit criteria [40]. The most predictive model estimates the highest values in correlation coefficient (r), coefficient of determination ($R^2$) and Adj $R^2$, and the lowest values in relative approximation error (RAE), mean absolute error (MAE), standard deviation ratio (SDratio), root mean square error (RMSE) and the coefficient of variation (CV, %), respectively. IBM-SPSS [34] software was also used for the Decision Trees' analysis

## Results

### Distribution of the qualitative traits

The frequency distribution of the colour traits of indigenous helmeted Guinea fowl is shown in Table 1. Agro-ecology significantly affected (P<0.05; P<0.01) all the six traits investigated. No definite pattern of variation in each class of the colour traits was observed among the three agro-ecological zones. Generally, the most frequent colour phenotype of helmeted Guinea fowl in Nigeria had pearl plumage colour (54.0%), pale red skin colour (94.2%), black shank colour (68.7%), brown eye colour (49.7%), white earlobe colour (54.8%) and helmet colour (72.6%). However, sex did not influence (P>0.05) all the six colour traits.

The frequencies of helmet shape and wattle size were significantly affected by agro-ecology (P<0.01) (Table 2). While most of the birds had single helmet shape (50.8%), which appeared to be more in the Sudano-Sahelian and Tropical Rainforest zones, wattle size did not follow a definite pattern. All the birds in the three agro-ecologies had wattle and were skeletally normal (P>0.01). However, sex had a significant effect (P<0.01) on helmet shape (where more females were single), wattle size (where that of males appeared larger), and wattle shape (where more females carried theirs flat).

### Biplot of the multiple correspondence analysis

The MCA revealed the association between the qualitative physical traits and agro-ecological zones in two dimensions (Fig 2). The first dimension was high and represented 93.2% of the deviation from independence while the second dimension signified 6.8% of the total variation based on the inertia. The agro-ecological zones were not clustered perfectly (as revealed bythe low inertia values of 0.168 and 0.012) considering the intermingling of some qualitative physical traits. This was more noticeable between birds in the Sudano-Sahelian and Tropical Rainforest zones. Therefore, discrimination of the traits appears very weak. However, on the right hand side of the biplot, peach black, orange and pale pink shank colour, dark skin colour, and red and slanted backward helmet seemed to be more associated with the Southern Guinea Savanna zone.

### The fixed effect of agro-ecology on biometric traits and morphological indices

The results of the univariate analysis revealed significant effect (P<0.05) of agro-ecology on the biometric traits and morphological indices of the guinea fowls [Medians (means in parentheses)] (Table 3). Overall, birds from the Southern Guinea Savanna zone had significantly higher values (P<0.05) for most zoometrical traits compared to their Sudano-Sahelian and Tropical Rainforest counterparts. However, the former and the latter were similar (P>0.05) in

**Table 1.** Frequency (%) of colour traits of indigenous helmeted Guinea fowl based on agro-ecology and sex.

| | | Agro-ecology | | | | | Sex | | | |
|---|---|---|---|---|---|---|---|---|---|---|
| | | Southern Guinea Savanna | Sudano-Sahelian | Tropical Rainforest | Total | Kruskall-Wallis test | Male | Female | Total | Kruskall-Wallis test |
| Traits | Class | n = 109 | n = 270 | n = 190 | n = 569 | | n = 158 | n = 411 | n = 569 | |
| Plumage colour | Pearl | 12.3 | 26.9 | 14.8 | 54.0 | 9.69** | 16.0 | 38.0 | 54.0 | 0.28ns |
| | Lavender | 1.1 | 1.2 | 2.8 | 5.1 | | 1.1 | 4.0 | 5.1 | |
| | Black | 1.9 | 7.4 | 6.3 | 15.6 | | 3.5 | 12.1 | 15.6 | |
| | White | 0.0 | 0.9 | 1.8 | 2.6 | | 0.7 | 1.9 | 2.6 | |
| | Brown | 3.9 | 5.3 | 4.4 | 13.5 | | 3.3 | 10.2 | 13.5 | |
| | Pied | 0.0 | 5.8 | 3.3 | 9.1 | | 3.2 | 6.0 | 9.1 | |
| | Total | | | | 100 | | | | 100 | |
| Skin colour | Dark | 5.8 | 0.0 | 0.0 | 5.8 | 147.58** | 1.6 | 4.2 | 5.8 | 0.004ns |
| | Pale red | 13.4 | 47.5 | 33.4 | 94.2 | | 26.2 | 68.0 | 94.2 | |
| | Total | | | | 100 | | | | 100 | |
| Shank colour | Orange | 0.5 | 0.0 | 0.0 | 0.5 | 25.61** | 0.0 | 0.5 | 0.5 | 2.16ns |
| | Black | 8.8 | 33.6 | 26.4 | 68.7 | | 18.1 | 50.6 | 68.7 | |
| | White | 0.0 | 3.2 | 2.8 | 6.0 | | 2.1 | 3.9 | 6.0 | |
| | Brown | 0.7 | 5.4 | 3.0 | 9.1 | | 3.2 | 6.0 | 9.1 | |
| | Peach Black | 7.2 | 4.4 | 1.2 | 12.8 | | 3.5 | 9.3 | 12.8 | |
| | Pale Pink | 1.4 | 0.0 | 0.0 | 1.4 | | 0.2 | 1.2 | 1.4 | |
| | Pale Red | 0.0 | 0.7 | 0.0 | 0.7 | | 0.2 | 0.5 | 0.7 | |
| | Red | 0.0 | 0.2 | 0.0 | 0.2 | | 0.2 | 0.0 | 0.2 | |
| | Pink With Black Spot | 0.2 | 0.0 | 0.0 | 0.2 | | 0.2 | 0.0 | 0.2 | |
| | Black-Orange | 0.4 | 0.0 | 0.0 | 0.4 | | 0.2 | 0.2 | 0.4 | |
| | Total | | | | 100 | | | | 100 | |
| Eye colour | White | 1.9 | 3.9 | 2.1 | 7.9 | 91.86** | 2.3 | 5.6 | 7.9 | 1.27ns |
| | Brown | 17.2 | 21.8 | 10.7 | 49.7 | | 14.8 | 35.0 | 49.7 | |
| | Pink | 0.0 | 0.9 | 0.0 | 0.9 | | 0.4 | 0.5 | 0.9 | |
| | Black | 0.0 | 20.4 | 20.0 | 40.4 | | 10.2 | 30.2 | 40.4 | |
| | Bluish | 0.0 | 0.5 | 0.5 | 1.1 | | 0.2 | 0.9 | 1.1 | |
| | Total | | | | 100 | | | | 100 | |
| Earlobe colour | White | 4.9 | 26.4 | 23.6 | 54.8 | 59.63** | 15.5 | 39.4 | 54.8 | 0.22ns |
| | Dirty White | 0.0 | 1.2 | 0.5 | 1.8 | | 0.5 | 1.2 | 1.8 | |
| | Bluish | 0.0 | 0.5 | 0.0 | 0.5 | | 0.2 | 0.4 | 0.5 | |
| | White Bluish | 0.0 | 1.1 | 0.5 | 1.6 | | 0.4 | 1.2 | 1.6 | |
| | Spotted | 4.9 | 8.4 | 4.0 | 17.4 | | 4.9 | 12.5 | 17.4 | |
| | Whitish Brown | 0.0 | 1.2 | 0.4 | 1.6 | | 0.4 | 1.2 | 1.6 | |
| | Brown | 6.0 | 7.9 | 4.2 | 18.1 | | 5.4 | 12.7 | 18.1 | |
| | Black | 0.2 | 0.2 | 0.0 | 0.4 | | 0.2 | 0.2 | 0.4 | |
| | Pale Pink | 1.9 | 0.0 | 0.0 | 1.9 | | 0.0 | 1.9 | 1.9 | |
| | Pink | 1.2 | 0.0 | 0.0 | 1.2 | | 0.4 | 0.9 | 1.2 | |
| | Purple | 0.0 | 0.5 | 0.2 | 0.7 | | 0.0 | 0.7 | 0.7 | |
| | Total | | | | 100 | | | | 100 | |
| Helmet colour | Purple | 0.0 | 0.2 | 0.0 | 0.2 | 53.17** | 0.0 | 0.2 | 0.2 | 0.03ns |
| | Brown | 9.3 | 37.6 | 25.7 | 72.6 | | 20.6 | 52.0 | 72.6 | |

*(Continued)*

**Table 1.** (Continued)

| | | Agro-ecology | | | | | Sex | | | |
|---|---|---|---|---|---|---|---|---|---|---|
| | | Southern Guinea Savanna | Sudano-Sahelian | Tropical Rainforest | Total | Kruskall-Wallis test | Male | Female | Total | Kruskall-Wallis test |
| Traits | Class | n = 109 | n = 270 | n = 190 | n = 569 | | n = 158 | n = 411 | n = 569 | |
| | Black | 2.3 | 6.3 | 6.2 | 14.8 | | 3.3 | 11.4 | 14.8 | |
| | Red | 7.6 | 2.8 | 1.6 | 12.0 | | 3.5 | 8.4 | 12.0 | |
| | Pink | 0.0 | 0.5 | 0.0 | 0.5 | | 0.4 | 0.2 | 0.5 | |
| | Total | | | | 100 | | | | 100 | |

n = No. of birds observed; * P<0.01; ns Not significant

all the biometric parameters. As regards morphological indices, Southern Guinea Savanna birds were more compact (120.00 vs. 110.00 vs. 107.69) but had lesser condition index (7.66 vs. 9.45 vs. 9.30) and lower long-leggedness (19.71 vs. 19.23 vs. 9.51) than those of Sudano-Sahelian and Tropical Rainforest agro-ecological zones.

## The fixed effect of sex on biometric traits and morphological indices irrespective of agro-ecologies

Across agro-ecological zones, sex significantly influenced (P<0.05) nine biometric traits and one morphological index [Medians (means in parentheses)] (Table 4). Male birds had higher

**Table 2.** Frequency (%) of helmet shape, wattle possession, size and shape including skeletal structure of indigenous helmeted Guinea fowl based on agro-ecology and sex.

| | | Agro-ecology | | | | | Sex | | | |
|---|---|---|---|---|---|---|---|---|---|---|
| | | Southern Guinea Savanna | Sudano-Sahelian | Tropical Rainforest | Total | Kruskall-Wallis test | Male | Female | Total | Kruskall-Wallis test |
| Traits | Class | n = 109 | n = 270 | n = 190 | n = 569 | | n = 158 | n = 411 | n = 569 | |
| Helmet shape | Slanted Backward | 13.0 | 5.3 | 4.0 | 22.3 | 43.61** | 6.2 | 16.2 | 22.3 | 94.57** |
| | Single | 0.2 | 29.3 | 21.3 | 50.8 | | 1.8 | 49.0 | 50.8 | |
| | Erect | 6.0 | 12.8 | 8.1 | 26.9 | | 19.9 | 7.0 | 26.9 | |
| | Total | | | | 100 | | | | 100 | |
| Wattle possession | Present | 19.2 | 47.5 | 33.4 | 100.0 | 0.00ns | 27.8 | 72.2 | 100 | 0.00ns |
| | Absent | 0.0 | 0.0 | 0.0 | 0.0 | | 0.0 | 0.0 | 0.0 | |
| | Total | | | | 100 | | | | 100 | |
| Wattle size | Large | 12.7 | 23.7 | 13.4 | 49.7 | 18.79** | 23.9 | 25.8 | 49.7 | 115.34** |
| | Small | 6.5 | 23.7 | 20.0 | 49.7 | | 3.9 | 46.4 | 50.3 | |
| | Total | | | | 100 | | | | 100 | |
| Wattle shape | Cupped | 5.1 | 13.9 | 8.8 | 27.8 | 0.47ns | 27.2 | 0.5 | 27.8 | 526.89** |
| | Flat | 14.1 | 33.0 | 24.3 | 71.4 | | 0.5 | 70.8 | 71.4 | |
| | Cupped Flat | 0.0 | 0.5 | 0.4 | 0.9 | | 0.0 | 0.9 | 0.9 | |
| | Total | | | | 100 | | | | 100 | |
| Skeletal structure | Normal | 19.2 | 47.5 | 33.4 | 100 | 0.00ns | 27.8 | 72.2 | 100 | 0.00ns |
| | Creeper | 0.0 | 0.0 | 0.0 | 0.0 | | 0.0 | 0.0 | 0.0 | |
| | Polydactyl | 0.0 | 0.0 | 0.0 | 0.0 | | 0.0 | 0.0 | 0.0 | |

n = No. of birds observed;

** Significant at P <0.01; ns Not significant

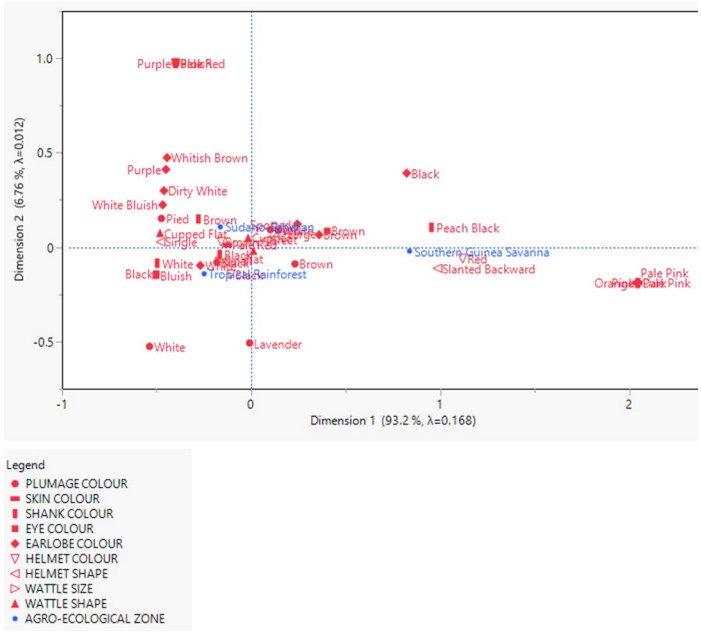

**Fig 2. A biplot showing the relationship between the qualitative physical traits and agro-ecological zones.**

**Table 3. Medians (means in parentheses) of biometric traits and morphological indices of indigenous helmeted Guinea fowls based on agro-ecology.**

| Traits | Agro-ecology | | | Kruskall-Wallis test | P-value |
|---|---|---|---|---|---|
| | Southern Guinea Savanna | Sudano-Sahelian | Tropical Rainforest | | |
| Body weight | 1.38 (1.49) | 1.40 (1.45) | 1.38 (1.41) | 3.49 | 0.175 |
| Head length | 4.30[b] (4.46) | 4.80[a] (4.74) | 4.80[a] (4.74) | 41.24 | 0.001 |
| Head thickness | 10.70[b] (10.80) | 11.00[a] (11.04) | 11.00[a] (11.01) | 13.84 | 0.001 |
| Helmet length | 2.20[a] (2.17) | 2.10[b] (1.96) | 2.10[b] (2.00) | 8.95 | 0.011 |
| Helmet width | 1.70[a] (1.63) | 1.20[b] (1.36) | 1.20[b] (1.31) | 56.72 | 0.001 |
| Wattle length | 2.30 (2.27) | 2.40 (2.32) | 2.30 (2.31) | 0.475 | 0.789 |
| Wattle width | 1.80[a] (1.83) | 1.40[b] (1.62) | 1.40[b] (1.52) | 45.75 | 0.001 |
| Neck length | 14.50[a] (13.97) | 11.00[b] (11.55) | 11.00[b] (11.20) | 134.85 | 0.001 |
| Neck circumference | 7.30 (7.89) | 7.00 (7.26) | 7.00 (7.18) | 5.68 | 0.058 |
| Wing length | 18.30[a] (17.69) | 14.60[b] (14.75) | 14.60[b] (14.67) | 133.14 | 0.001 |
| Wing Span | 39.10[a] (38.68) | 34.50[b] (35.32) | 34.20[b] (35.03) | 129.61 | 0.001 |
| Body length | 38.40[a] (38.28) | 34.50[b] (36.05) | 34.40[b] (35.53) | 67.89 | 0.001 |
| Trunk Length | 25.60 (26.46) | 26.00 (26.02) | 26.00 (25.94) | 0.39 | 0.822 |
| Keel length | 11.00[a] (11.31) | 11.00[ab] (10.95) | 11.00[b] (10.77) | 12.39 | 0.002 |
| Chest circumference | 31.00[a] (32.05) | 29.00[b] (29.39) | 27.60[b] (28.72) | 52.58 | 0.001 |
| Thigh length | 11.00[a] (11.51) | 10.20[b] (10.79) | 10.00[b] (10.65) | 38.92 | 0.001 |
| Shank length | 7.30[a] (7.35) | 7.00[b] (7.03) | 7.00[b] (7.00) | 27.16 | 0.001 |
| Shank thickness | 4.00[b] (4.02) | 5.40[a] (5.42) | 5.40[a] (5.39) | 156.26 | 0.001 |
| Massiveness | 5.40 (5.66) | 5.19 (5.61) | 5.17 (5.48) | 2.85 | 0.240 |
| Compactness | 120.00[a] (121.71) | 110.00[b] (113.35) | 107.69[b] (111.06) | 48.59 | 0.001 |
| Long-leggedness | 19.23[b] (19.24) | 19.71[a] (19.74) | 19.51[a] (19.94) | 13.06 | 0.001 |
| Condition index | 7.66[b] (8.68) | 9.45[a] (9.82) | 9.30[a] (9.63) | 40.70 | 0.001 |

Mean ranks within rows with P<0.05 are significantly different

**Table 4. Medians (means in parentheses) of biometric traits and morphological indices of indigenous helmeted Guinea fowls based on sex.**

| Traits | Sex | | Kruskall-Wallis test | P-value |
|---|---|---|---|---|
| | Male | Female | | |
| Body weight (kg) | 1.40 (1.49) | 1.40 (1.43) | 6.19 | 0.013 |
| Head length (cm) | 4.70 (4.71) | 4.70 (4.68) | 0.16 | 0.686 |
| Head thickness (cm) | 11.00 (11.12) | 11.00 (10.94) | 10.40 | 0.001 |
| Helmet length (cm) | 2.15 (2.07) | 2.10 (1.99) | 0.983 | 0.322 |
| Helmet width (cm) | 1.40 (1.43) | 1.30 (1.38) | 2.13 | 0.144 |
| Wattle length (cm) | 2.40 (2.35) | 2.30 (2.29) | 3.00 | 0.083 |
| Wattle width (cm) | 1.50 (1.73) | 1.40 (1.58) | 11.58 | 0.001 |
| Neck length (cm) | 11.00 (11.96) | 11.00 (11.87) | 0.23 | 0.630 |
| Neck circumference (cm) | 7.00 (7.38) | 7.00 (7.35) | 4.04 | 0.044 |
| Wing length (cm) | 15.00 (15.57) | 14.60 (15.18) | 9.59 | 0.002 |
| Wing Span (cm) | 35.00 (36.25) | 34.60 (35.72) | 1.25 | 0.263 |
| Body length (cm) | 35.00 (37.11) | 35.00 (35.99) | 8.16 | 0.004 |
| Trunk Length (cm) | 26.25 (26.50) | 26.00 (25.91) | 8.66 | 0.003 |
| Keel length (cm) | 11.00 (11.10) | 11.00 (10.91) | 2.96 | 0.086 |
| Chest circumference (cm) | 30.00 (30.31) | 29.00 (29.44) | 8.55 | 0.003 |
| Thigh length (cm) | 11.00 (11.07) | 10.20 (10.81) | 10.98 | 0.001 |
| Shank length (cm) | 7.00 (7.12) | 7.00 (7.06) | 2.05 | 0.152 |
| Shank thickness (cm) | 5.20 (5.27) | 5.00 (5.09) | 3.25 | 0.071 |
| Massiveness | 5.19 (5.67) | 5.18 (5.54) | 0.30 | 0.581 |
| Compactness | 111.32 (114.93) | 110.00 (113.90) | 0.71 | 0.399 |
| Long-leggedness | 19.43 (19.40) | 19. 44 (19.83) | 4.21 | 0.040 |
| Condition index | 9.36 (9.64) | 9.26 (9.50) | 1.19 | 0.276 |

Mean ranks within rows with P <0.05 are significantly different

body weight, head thickness, wattle width, neck circumference, wing length, body length, trunk length, chest circumference and thigh length. However, female birds had higher long-leggedness (19. 44 vs. 19.43) compared to males.

## The fixed effect of sexes within agro-ecologies on biometric traits and morphological indices

The effect of sexes within agro-ecologies had significant effect (P<0.05) on some biometric traits and morphological indices in two out of the three agro-ecological zones [Medians (means in parentheses)] (Table 5). In the Sudano-Sahelian zone, the body weight of males (1.45) was higher than that of the females (1.40) likewise head thickness, wattle width, wing length, body length, trunk length, keel length, chest circumference, thigh length and condition index. However, the female birds had higher long-leggedness (20.29 vs. 19.43) than their male counterparts. Male birds also had higher wattle length (2.80 vs. 2.30), wattle width (2.00 vs. 1.70), neck circumference (8.00 vs. 7.00), body length (39.10 vs. 38.05), trunk length (26.80 vs. 25.50), thigh length (11.40 vs. 11.00) and shank thickness (4.10 vs. 3.90) in the Southern Guinea Savanna zone.

## Spatial representation of birds

Based on Wilks' Lambda (0.326–0.663) and F statistics (41.855–143.662) (Table 6), wing length, shank thickness, massiveness, neck circumference, head thickness, condition index,

**Table 5. Medians (means in parentheses) of biometric traits and morphological indices of indigenous helmeted Guinea fowls of sexes within agro-ecologies.**

| Traits | Southern Guinea Savanna | | | Sudano-Sahelian | | | Tropical Rainforest | | |
|---|---|---|---|---|---|---|---|---|---|
| | Male | Female | $X^2$ | Male | Female | $X^2$ | Male | Female | $X^2$ |
| BW | 1.41 (1.48) | 1.37 (1.50) | $0.001^{ns}$ | 1.45 (1.53) | 1.40 (1.42) | $12.92^{**}$ | 1.35 (1.43) | 1.40 (1.41) | $0.903^{ns}$ |
| HDL | 4.50 (4.54) | 4.20 (4.44) | $1.78^{ns}$ | 4.90 (4.79) | 4.80 (4.72) | $0.84^{ns}$ | 4.60 (4.67) | 4.80 (4.76) | $0.165^{ns}$ |
| HDT | 10.80 (10.93) | 10.60 (10.75) | $1.92^{ns}$ | 11.00 (11.20) | 11.00 (10.98) | $8.87^{**}$ | 11.00 (11.10) | 11.00 (10.98) | $0.211^{ns}$ |
| HL | 2.40 (2.57) | 2.20 (2.04) | $3.80^{ns}$ | 2.05 (1.96) | 2.10 (1.96) | $0.07^{ns}$ | 2.10 (1.98) | 2.10 (2.00) | $0.783^{ns}$ |
| HW | 1.70 (1.66) | 1.70 (1.61) | $0.06^{ns}$ | 1.40 (1.40) | 1.20 (1.35) | $1.27^{ns}$ | 1.20 (1.38) | 1.20 (1.29) | $0.238^{ns}$ |
| WL | 2.80 (2.53) | 2.30 (2.19) | $9.51^{**}$ | 2.40 (2.34) | 2.20 (2.32) | $0.41^{ns}$ | 2.20 (2.28) | 2.40 (2.32) | $0.598^{ns}$ |
| WW | 2.00 (1.99) | 1.70 (1.77) | $7.19^{**}$ | 1.50 (1.76) | 1.40 (1.56) | $9.21^{**}$ | 1.40 (1.55) | 1.30 (1.51) | $0.380^{ns}$ |
| NL | 14.50 (14.14) | 14.50 (13.92) | $0.13^{ns}$ | 11.00 (11.64) | 11.00 (11.52) | $1.15^{ns}$ | 11.00 (11.32) | 11.00 (11.15) | $0.929^{ns}$ |
| NC | 8.00 (8.29) | 7.00 (7.76) | $12.02^{**}$ | 7.00 (7.10) | 7.00 (7.33) | $0.97^{ns}$ | 7.00 (7.32) | 7.00 (7.13) | $0.147^{ns}$ |
| WGL | 19.70 (18.26) | 18.15 (17.50) | $2.17^{ns}$ | 15.00 (15.11) | 14.60 (14.60) | $15.98^{**}$ | 14.60 (14.86) | 14.60 (14.60) | $0.088^{ns}$ |
| WGS | 40.00 (39.28) | 39.00 (38.49) | $1.03^{ns}$ | 34.60 (35.89) | 34.40 (35.07) | $1.76^{ns}$ | 34.20 (35.22) | 34.20 (34.96) | $0.594^{ns}$ |
| BL | 39.10 (39.40) | 38.05 (37.91) | $5.83^{*}$ | 35.00 (37.11) | 33.10 (35.60) | $11.29^{**}$ | 35.00 (35.90) | 33.40 (35.39) | $0.472^{ns}$ |
| TRL | 26.80 (27.52) | 25.50 (26.11) | $5.68^{*}$ | 26.25 (26.44) | 26.00 (25.84) | $3.91^{*}$ | 26.00 (26.06) | 26.00 (25.89) | $0.362^{ns}$ |
| KL | 10.90 (11.07) | 11.20 (11.39) | $1.49^{ns}$ | 11.00 (11.20) | 11.00 (10.85) | $4.74^{*}$ | 11.00 (10.96) | 11.00 (10.70) | $0.086^{ns}$ |
| CC | 32.00 (32.60) | 31.00 (31.87) | $0.47^{ns}$ | 30.00 (30.32) | 27.20 (29.00) | $9.82^{**}$ | 29.00 (29.07) | 27.20 (28.59) | $0.267^{ns}$ |
| TL | 11.40 (11.89) | 11.00 (11.38) | $5.57^{*}$ | 10.50 (11.00) | 10.20 (10.70) | $7.05^{**}$ | 10.30 (10.75) | 10.00 (10.62) | $0.139^{ns}$ |
| SL | 7.50 (7.46) | 7.25 (7.31) | $3.08^{ns}$ | 7.00 (7.08) | 7.00 (7.01) | $0.33^{ns}$ | 7.00 (7.01) | 7.00 (7.00) | $0.602^{ns}$ |
| ST | 4.10 (4.25) | 3.90 (3.95) | $11.52^{**}$ | 5.90 (5.54) | 5.40 (5.36) | $1.50^{ns}$ | 5.40 (5.38) | 5.40 (5.39) | $0.840^{ns}$ |
| MS | 5.19 (5.40) | 5.44 (5.74) | $1.41^{ns}$ | 5.56 (5.86) | 5.18 (5.51) | $2.88^{ns}$ | 5.17 (5.51) | 5.17 (5.47) | $0.782^{ns}$ |
| CP | 117.24 (119.09) | 121.67 (122.57) | $0.98^{ns}$ | 110.71 (115.44) | 110.00 (112.47) | $1.23^{ns}$ | 110.42 (111.92) | 107.69 (110.75) | $0.436^{ns}$ |
| LL | 18.97 (19.01) | 19.23 (19.32) | $0.30^{ns}$ | 19.43(19.31) | 20.29 (19.93) | $5.96^{*}$ | 19.71 (19.75) | 19.51 (20.01) | $0.784^{ns}$ |
| CI | 7.25 (8.27) | 7.70 (8.81) | $1.27^{ns}$ | 9.72(10.14) | 9.35 (9.69) | $5.70^{*}$ | 9.25 (9.59) | 9.33 (9.64) | $0.592^{ns}$ |

BW, body weight (kg); HDL, head length (cm); HDT, head thickness (cm); HL, helmet length (cm); HW, helmet width (cm); WL (cm), wattle length (cm); WW, wattle width (cm); NL, neck length (cm); NC, neck circumference (cm); WGL, wing length (cm); WGS, wing span (cm); BL, body length (cm); TRL, trunk length (cm); KL, keel length (cm); CC, chest circumference (cm); TL, thigh length (cm); SL, shank length (cm); ST, shank thickness (cm); MS, massiveness; CP, compactness; LL, long-leggedness; CI, condition index.

$X^2$, Kruskal-Wallis H test value

*, **, Significant at P <0.05 and P <0.01, respectively; ns, Not significant

Mean ranks within rows with P <0.05 are significantly different for sexes within each agro-ecological zone.

long-leggedness, neck length, thigh length and wattle length were the significant (P<0.001) parameters of importance to separate birds in the Southern Guinea Savanna, Sudano-Sahelian

**Table 6. Traits of importance in the discriminant analysis to separate birds in the three agro-ecological zones.**

| Traits | Wilk's Lambda | F-value | P-Level | Tolerance |
|---|---|---|---|---|
| Wing length | 0.663 | 143.662 | 0.001 | 1.000 |
| Shank thickness | 0.562 | 94.269 | 0.001 | 0.990 |
| Massiveness | 0.506 | 76.328 | 0.001 | 0.764 |
| Neck circumference | 0.465 | 65.712 | 0.001 | 0.925 |
| Head thickness | 0.420 | 61.016 | 0.001 | 0.785 |
| Condition index | 0.387 | 56.756 | 0.001 | 0.131 |
| Long-leggedness | 0.362 | 52.956 | 0.001 | 0.726 |
| Neck length | 0.343 | 49.427 | 0.001 | 0.729 |
| Thigh length | 0.332 | 45.638 | 0.001 | 0.577 |
| Wattle length | 0.326 | 41.855 | 0.001 | 0.582 |

and Tropical Rainforest zones. However, there was considerable spatial intermixing of the bio-metric traits largely observed between birds in the Sudano-Sahelian and Tropical Rainforest zones (Fig 3). The predicted group membership of the three agro-ecological zones is shown in Table 7. The classification results showed that 88.1, 51.9 and 55.8% of birds in the Southern Guinea Savanna, Sudano-Sahelian and Tropical Rainforest zones, respectively were correctly assigned to their distinct groups. The three respective group cases were 57.1% cross-validated.

## Contributions to variation and loadings of variables on the principal components

The result of CATPCA revealed the extraction of two principal components (PCs) which explained 42.1% of the variation in the dataset (Table 8). The first PC (Eigenvalue = 8.386) explained 27.1% of the total variance and was greatly influenced by body length (0.832), body weight (0.830), compactness (0.812), massiveness (0.810), helmet length (-0.748), wattle width (0.755), chest circumference (0.741), wattle length (-0.730), helmet width (0.723), thigh length (0.713), shank length (0.642), long-leggedness (-0.616), head thickness (0.608), condition index (0.532), and neck circumference (0.391) (Fig 4). Agro-ecology (-0.751) was more associ-ated with the second PC (Eigenvalue = 4.652) which accounted for 15.0% of the total variation and had its loadings for wing length (0.754), skin colour (-0.679), neck length (0.647), head length (-0.634), wing span (0.632), eye colour (-0.504), shank thickness (-0.490), helmet colour (0.467), helmet shape (-0.419), earlobe colour (0.390), wattle size (-0.359), plumage colour (-0.254), keel length (0.246), shank colour (0.207), and trunk length (0.126). Wattle shape had equal loading for PC1 and PC2 (-0.088). However, the contributions of sex of birds to both PC1 (-0.094) and PC2 (-0.079) in terms of loadings were negligible. The high Cronbach's alpha value of 0.954 indicates the reliability of the CATPCA.

## Decision trees of the data mining

The tree diagram of the CHAID algorithm is shown in Fig 5. Seven terminal nodes (Nodes 1, 2, 3, 5, 6, 7 and 8) were formed. The root node (Node 0) showed the descriptive statistics of the birds in the three agro-ecological zones. The Chi-squared-based branch and node distribution

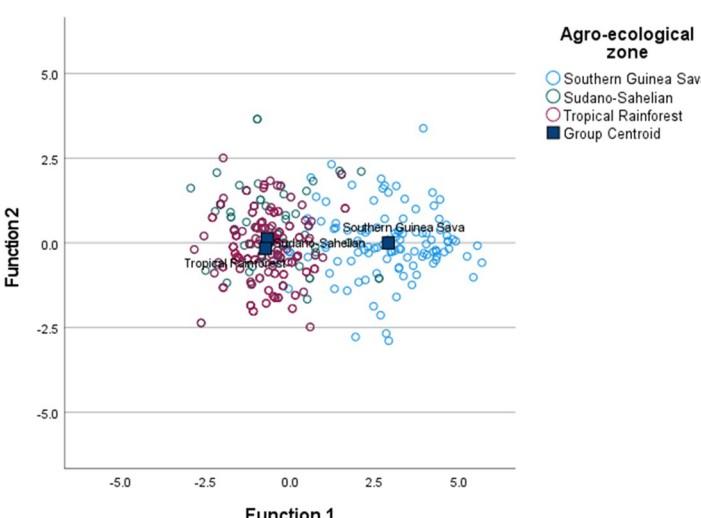

**Fig 3. Canonical discriminant function illustrating the distribution of the Guinea fowls among the agro-ecological zones.**

**Table 7.  Assignment of birds to the three agro-ecological zones.**

| | | Predicted group membership | | | |
|---|---|---|---|---|---|
| | Agro-ecology | Southern Guinea Savanna | Sudano-Sahelian | Tropical Rainforest | Total |
| Original count | Southern Guinea Savanna | 96 | 8 | 5 | 109 |
| | Sudano-Sahelian | 12 | 140 | 118 | 270 |
| | Tropical Rainforest | 3 | 81 | 106 | 190 |
| % | Southern Guinea Savanna | 88.1 | 7.3 | 4.6 | 100.0 |
| | Sudano-Sahelian | 4.4 | 51.9 | 43.7 | 100.0 |
| | Tropical Rainforest | 1.6 | 42.6 | 55.8 | 100.0 |
| Cross-validated count | Southern Guinea Savanna | 93 | 9 | 7 | 109 |
| | Sudano-Sahelian | 12 | 132 | 126 | 270 |
| | Tropical Rainforest | 3 | 87 | 100 | 190 |
| % | Southern Guinea Savanna | 85.3 | 8.3 | 6.4 | 100.0 |
| | Sudano-Sahelian | 4.4 | 48.9 | 46.7 | 100.0 |
| | Tropical Rainforest | 1.6 | 45.8 | 52.6 | 100.0 |

60.1% of original grouped cases correctly classified.

57.1% of cross-validated grouped cases correctly classified.

revealed that wing length was the variable of utmost importance in assigning the birds into their respective agro-ecological zone followed by eye colour. Wing length (>18.10 cm) only was significantly (P<0.001) sufficient to discriminate between birds of the Southern Guinea Savanna and those of Sudano-Sahelian and Tropical Rainforest zones. However, wing length (14.80–15.50 cm) together with eye colour provided a better differentiation of the Sudano-Sahelian and Tropical Rainforest zones. While birds from the former had mostly brown and pink eye colour, the later were associated mostly with white, black and bluish eye colour. It was observed that 52.3, 86.7, and 20.5% of birds in the Southern Guinea Savanna, Sudano-Sahelian and Tropical Rainforest zones, respectively, were correctly assigned to their distinct agro-eco-logical zone with an average accuracy rate of 58.0% (Table 9). The r, $R^2$, Adj $R^2$, RAE, MAE, SDratio, RMSE and CV (%) values were 0.481, 0.231, 0.230, 0.287, 0.111, 0.898, 0.648, and 29.83, respectively.

The Exhaustive CHAID decision tree formed seven terminal nodes (Nodes 3, 4, 5, 6, 8, 9 and 10) (Fig 6). Here, wing length (>18.10 cm) was also the best single discriminant variable (P<0.001) to distinguish birds in the three agro-ecological zones. In contrast to what was obtained under CHAID, body length and eye colour were the two additional variables to dif-ferentiate the populations. Wing length (14.80–15.50 cm), body length (< = 35.00 cm) and eye colour permitted a better separation of the Sudano-Sahelian from Tropical Rainforest birds. Unlike what was observed in CHAID, birds from the former had mostly brown, white and pink eye colours while the later were characterized by black as well as bluish eye colour. In this model, 52.3, 77.4 and 37.9% of birds in the Southern Guinea Savanna, Sudano-Sahelian and Tropical Rainforest zones, respectively, were correctly assigned to their distinct agro-ecological zone with an average accuracy rate of 59.4% (Table 10). The r, $R^2$, Adj $R^2$, RAE, MAE, SDratio, RMSE and CV (%) values were 0.520, 0.270, 0.268, 0.282, 0.009, 0.896, 0.637, and 29.765, respectively.

## Discussion

Phenotypic variation of local animal resources indicates a genetic diversity that may be worth conserving for future uses while better understanding of the external features helps to facilitate

**Table 8. Eigenvalue and the contribution of each qualitative and quantitative trait to the total variation in the principal components.**

| Traits | PC1 | PC2 | Total |
|---|---|---|---|
| Plumage colour | 0.002 | 0.065 | 0.066 |
| Skin colour | 0.010 | 0.461 | 0.471 |
| Shank colour | 0.029 | 0.043 | 0.072 |
| Eye colour | 0.011 | 0.254 | 0.265 |
| Earlobe colour | 0.030 | 0.152 | 0.182 |
| Helmet colour | 0.003 | 0.218 | 0.221 |
| Helmet shape | 0.003 | 0.175 | 0.178 |
| Wattle size | 0.001 | 0.129 | 0.130 |
| Wattle shape | 0.008 | 0.008 | 0.015 |
| Body weight | 0.689 | 0.013 | 0.702 |
| Head length | 0.035 | 0.402 | 0.437 |
| Head thickness | 0.370 | 0.024 | 0.394 |
| Helmet length | 0.560 | 0.002 | 0.562 |
| Helmet width | 0.522 | 0.119 | 0.641 |
| Wattle length | 0.533 | 0.001 | 0.535 |
| Wattle width | 0.570 | 0.108 | 0.678 |
| Neck length | 0.255 | 0.419 | 0.674 |
| Neck circumference | 0.153 | 0.042 | 0.195 |
| Wing length | 0.125 | 0.569 | 0.694 |
| Wing Span | 0.053 | 0.400 | 0.452 |
| Body length | 0.692 | 0.036 | 0.728 |
| Trunk Length | 0.008 | 0.016 | 0.023 |
| Keel length | 0.028 | 0.061 | 0.089 |
| Chest circumference | 0.549 | 0.074 | 0.623 |
| Thigh length | 0.508 | 0.021 | 0.529 |
| Shank length | 0.413 | 0.016 | 0.429 |
| Shank thickness | 0.246 | 0.240 | 0.486 |
| Massiveness | 0.656 | 0.014 | 0.671 |
| Compactness | 0.660 | 0.059 | 0.719 |
| Long-leggedness | 0.380 | 0.090 | 0.470 |
| Condition index | 0.283 | 0.425 | 0.708 |
| Agro-ecology[b] | 0.016 | 0.561 | 0.577 |
| Sex[b] | 0.009 | 0.006 | 0.015 |
| Eigenvalue | 8.386 | 4.652 | 13.038 |
| % of Variance | 27.052 | 15.006 | 42.059 |

b = Supplementary variable.

the implementation of conservation policies aimed to ensure local resources survival [15]. Morphometric and phaneroptic approaches may be fundamental in the management of poultry, considering the fact that they are fast and economically profitable [37]. The preponderance of more female birds in the present study could be attributed to the fact that smallholder poultry farmers normally keep more hens for the purpose of procreation, whereas the cocks are mostly slaughtered for consumption or sold to generate family income. We observed four major plumage colours (Pearl, Black, Brown and Pied). The varying colour patterns could be an indication that there are no pure genotypes of Guinea fowl in Nigeria as there are no

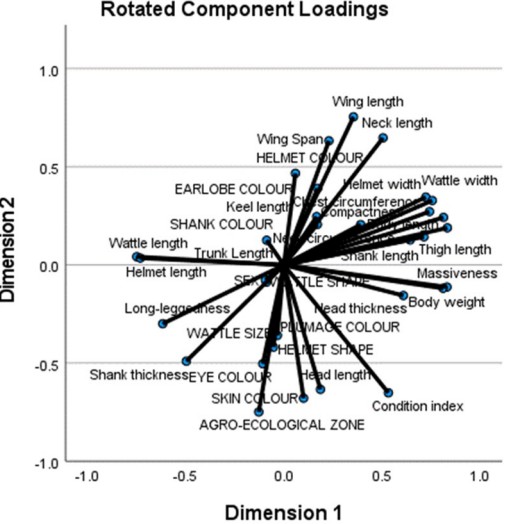

**Fig 4. Individual biometric traits, morphological indices and qualitative physical traits loadings on the principal components.**

records of selective breeding of the indigenous stock birds. However, the colour patterns were somehow different from the dominant Pearl, Lavender, Black and White variations earlier reported in the country [42, 43]. The slight variation may be occasioned by sampling coverage. In a similar study in Ghana, Agbolosu et al. [21] found that the predominant plumage colour was pearl grey colour (43.7%), whereas Traore et al. [22] reported pied plumage colour (42.76%) as the most frequent colour in the provinces of Burkina Faso. The Nigerian birds shared brown eye colour (57.0%) with those of Atakora (Mountainous) dry savannah zone in Togo [44] and black shank colour with those of Kenya (95.6%) [16], Sudanian and Sudano-Guinean zones in Benin [45]. Colour polymorphism defies evolutionary expectations as a

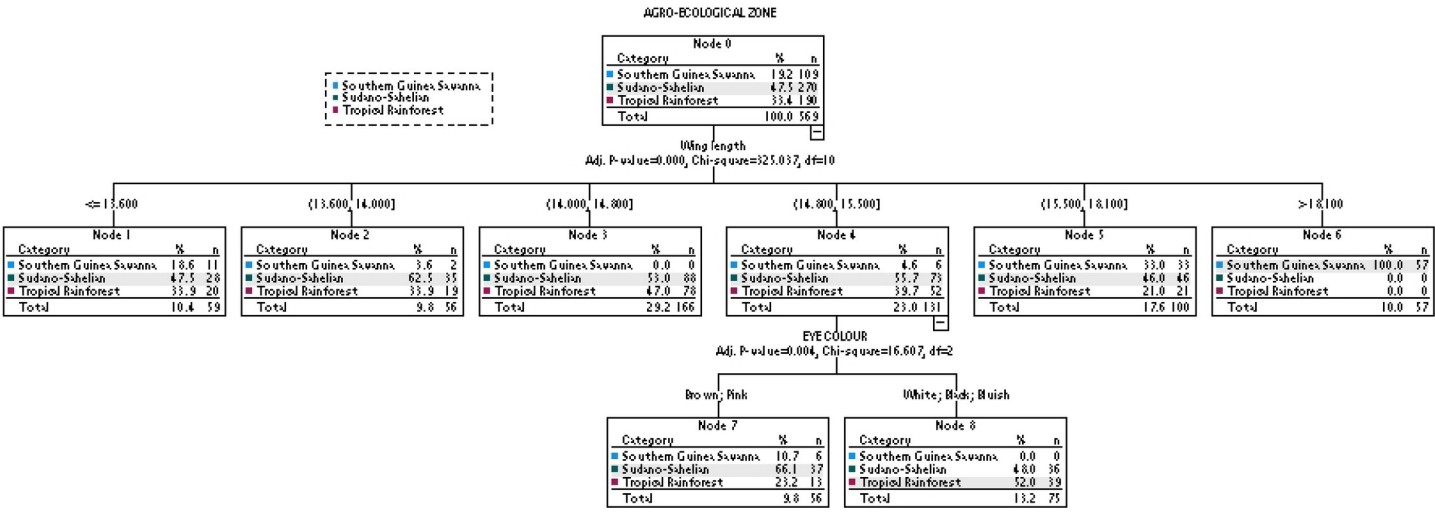

**Fig 5. Association between the agro-ecologies and the phenotypic traits using CHAID.**

**Table 9. The classification matrix of birds in the three agro-ecological zones based on CHAID model.**

| | | Predicted group membership | | | |
|---|---|---|---|---|---|
| | Agro-ecology | Southern Guinea Savanna | Sudano-Sahelian | Tropical Rainforest | % of correctly classified |
| Observed group membership | Southern Guinea Savanna | 57 | 52 | 0 | 52.3 |
| | Sudano-Sahelian | 0 | 234 | 36 | 86.7 |
| | Tropical Rainforest | 0 | 151 | 39 | 20.5 |
| | Overall % | 10.0 | 76.8 | 13.2 | 58.0 |

single species may maintain a striking phenotypic variation [46]. The present variant phenotypes may be due to polymorphism [47] and might have evolved in local Guinea fowls as adaptive measures for survival under varied environmental conditions. According to Getachew et al. [48], sustainable livestock production in the tropics requires adaptive genotypes which can withstand the undesirable effects of climate change and ensure optimal performance of the birds. In another study on a different species, Nigenda-Morales et al. [49] reported that the overall fitness of individuals in their environments may be affected by colour while Gong et al. [50] considered colour variation as an environmental indicator, which provides clues for the study of population genetics and biogeography. The preponderance of Pearl plumage colour in our study may also be attributed to farmers' preference, which is congruous with the submission of Vignal et al. [4] that prevalence of a particular colour could be attached to social-cultural value without any proven relationship with a biological function. This was buttressed by the report of González Ariza et al. [37] that certain qualitative physical traits may be associated with consumers' trends and their cultural preferences. Our findings on helmet shape are

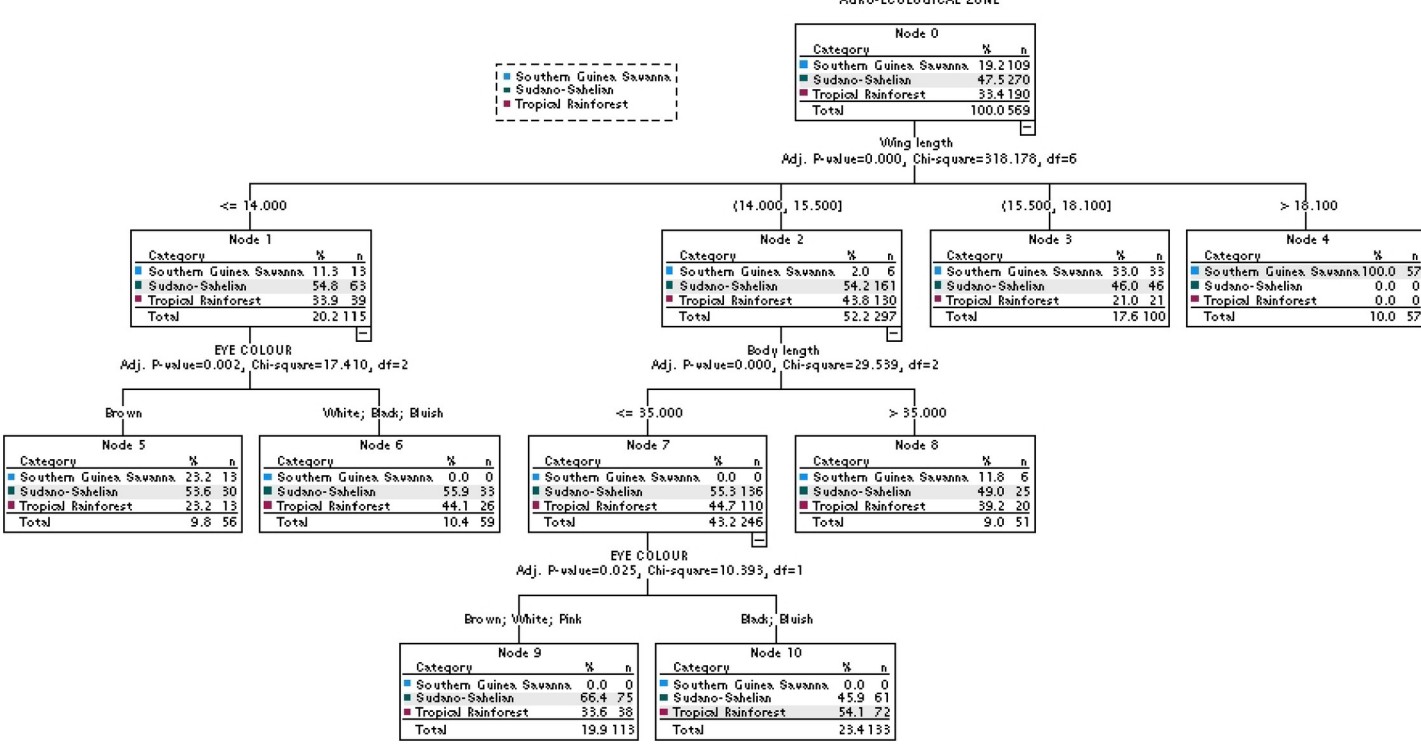

**Fig 6. The association between the agro-ecologies and the phenotypic traits using Exhaustive CHAID.**

**Table 10. The classification matrix of birds in the three agro-ecological zones based on Exhaustive CHAID model.**

| | | Predicted group membership | | | |
|---|---|---|---|---|---|
| | Agro-ecology | Southern Guinea Savanna | Sudano-Sahelian | Tropical Rainforest | % of correctly classified |
| Observed group membership | Southern Guinea Savanna | 57 | 52 | 0 | 52.3 |
| | Sudano-Sahelian | 0 | 209 | 61 | 77.4 |
| | Tropical Rainforest | 0 | 118 | 72 | 37.9 |
| | Overall % | 10.0 | 66.6 | 23.4 | 59.4 |

in agreement with the report on indigenous Guinea fowls in Ghanian where single shape (42.70%) predominated. The current observation on helmet shape where more females exhibited single shape is congruous withthe submission of Angst et al. [51] that females have bony helmet more compact dorsoventrally while the males have taller helmet, with a more complex shape including curvature of the posterior part along the dorsoventral axis. Similarly, Agbolosu et al. [21] reported that helmet shape is more pronounced in males than females. The observation on wattle is in consonance with the findings of Umosen et al. [52] who stated that, on the average, females had small wattle which was mostly flat.

In order to ascertain the genetic purity of the birds, the MCA result did not give a perfect clustering of the birds as phenotypic homogeneity of the Guinea fowl populations was evident in Sudano-Sahelian and Tropical Rainforest birds. This is in spite of the wide geographical distance and varying environmental conditions between the two zones. This suggests that colour traits alone might not be enough to distinguish between the three agro-ecological zones. Similar submission was made by Traore et al. [22] where, in spite of the enormous environmental differences, there was morphological homogeneity in qualitative traits in Guinea fowls in Burkina Faso. Brown et al. [36] also observed limited phenotypic and genetic diversity in local Guinea fowls in northern Ghana.

Univariate analysis revealed significant differences among zones for most biometric traits and calculated body indices, suggesting the possible influence of these zones on the evolutionary adaptation of the Guinea fowl population. However, there was no clear cut pattern in the biometric traits and morphological indices especially of the Sudano-Sahelian and Tropical Rainforest birds. The body weights of the present study are comparable to the 1.40 kg reported by Orounladji et al. [45] for indigenous Guinea fowls in a Sudanian zone in Benin. They are however, higher than the range 1.08–1.33 kg reported for adult Guinea fowl (*Numida meleagris*) in a humid zone of southern Nigeria [53] and 1.275 kg obtained in Zimbabwe [54]. Nevertheless, the indigenous birds are smaller in size when compared to their exotic counterparts. While Agwunobi and Ekpenyong [55] obtained a live weight of 1.5 kg for 'Golden Sovereign' Guinea fowl broiler strain under tropical conditions of Nigeria, Batkowska et al. [56] found a range of 2166 ± 42.5–2291 ± 46.9 kg for French commercial set. The differences may be attributed to genetics, age, physiological stage, location and management systems employed by the keepers. According to Ahiagbe et al. [57], genetic make-up and management practices could affect the growth traits of Guinea fowls. Exotic Guinea fowls are products of many years of robust selection and breeding [58, 59]. Therefore, it is possible that crossbreeding between the indigenous and exotic will result in birds of high genetic superiority in terms of meat yield and quality, egg production and adaptation. Sexual dimorphism provides insight into the sexual- and natural-selection pressures being experienced by male and female animals of different species [60]. At inter-population level, especially with some biometric traits, sexual dimorphism in the present study favoured male animals. This concurs with the established literature that males generally possess larger body sizes than females in normal sexual size dimorphism in birds [61]. The differential rate and duration of growth by the sexes may be responsible for the

present observations. Also, high rate of breeding in the populations could be another contributing factor to sexually dimorphic traits [62], as the birds have not been selected for the purpose of classical breeding. As obtained in the current study, Dudusola et al. [53] found male dominance in thigh length, body length, wing length, wing span, wattle length and chest circumference in Nigeria while Brown et al. [36] reported longer body and shank length including wingspan in indigenous Guinea fowl in Ghana. In a related study on domestic chicken, Toalombo Vargas et al. [63] reported longer thigh length in male birds.

The canonical discriminant analysis showed high level of admixture especially between the Sudano-Sahelian and Tropical Rainforest populations. It could, therefore, be reported that the Guinea fowls in Nigeria are unselected and largely of mixed populations. Northern Nigeria is the traditional home of indigenous helmeted guinea fowls in the country [64]. Considering the geographical proximity of the Southern Guinea Savanna and Sudano-Sahelian zones, one would have expected considerable intermixing of the guinea fowl populations. However, the reverse was observed in the present study as the intermingling between the birds in the Sudano-Sahelian and Tropical Rainforest zones was higher which could partly be due to transhumance especially by herders. The herders (mainly cattle rearers) from the northern parts of the country do move to the southern parts in search of natural pastures during the dry season. When they do so, they tend to carry along all their animals to their new locations. In that process, there is the possibility of exchange of birds between the settlers and their hosts. Such livestock mobility, which is seen as a means to an end [65] could have shaped poultry distribution pattern. Suffice to say that the guinea fowl (*Numida meleagris*) population of Tropical Rainforest is an ecotype of the Sudano-Sahelian; which is quite different from *Numida ptilorhycha* that is indigenous to the deciduous rain forest zone of southern Nigeria [66]. This assertion is corroborated by the reports of Ayorinde [67] and Obike et al. [68] who observed that *Numida meleagris*, domiciled in the north was spreading to other smallholder farming areas. In a related study, Whannou et al. [69] submitted that the mobility of herders could engender genetic introgression, thereby affecting animal genetic diversity. Another possible factor that could have contributed to the genetic erosion is inter-regional trade. It appears such live animal trade seemed to be more between livestock marketers in the Tropical Rainforest and Sudano-Sahelian zones than their Southern Guinea Savanna counterparts. According to Benton et al. [70], market dynamics in one location could drive biodiversity-damaging practices in other locations. In another study, Valerio et al. [71] highlighted the relevance of cross-border ties suggesting that markets play distinct structural roles in understanding animal movement patterns.

The results of CATPCA showed that some levels of separation of the Guinea fowls can be obtained based on agro-ecology which was more associated with the second principal component. The body parameters of importance in this component are wing length, skin colour, neck length, head length, wing span, eye colour, shank thickness, helmet colour, helmet shape, earlobe colour, wattle size, plumage colour, keel length, shank colour and trunk length. These parameters describe more of shape and colour of the guinea fowls. However, these differences in biometric traits and morphological indices based on agro-ecology were weak due to the fact that the second principal component could only account for 15.0% of the total variation. The use of CATPCA in assigning birds to their genetic groups had earlier been reported [33].

The decision tree results revealed that the guinea fowls from the Southern Guinea Savanna, Sudano-Sahelian and Tropical Rainforest zones could to a relative extent be separated using wing length, body length, and eye colour. However, the average accuracy rate of 58.0% (CHAID) and 59.4% (Exhaustive CHAID) obtained in this study indicated that 42 and 31.6% of the birds were wrongly classified. The implication of this is that there is a form of intermixing of the birds in the three agro-ecological zones. Both wing and body lengths are skeletal

parameters that are not influenced by body condition, thereby providing good estimates of overall body size of the birds. It is possible that both traits are under similar selection pressure [72]. The importance of morphometric traits in population stratification has also been stressed in other avian species [73, 74].

When all the algorithms used in this study are jointly considered, it could be said that the Guinea fowls from the Southern Guinea Savanna, Sudano-Sahelian and Tropical Rainforest zones of Nigeria are more homogeneous than heterogeneous in terms of the investigated qualitative physical traits, biometric traits and morphological indices. The biological implication of this is that elite birds from the three agro-ecological zones could be selected for the purpose of pure breeding or crossbreeding with their more productive exotic counterparts. This is beneficial considering the fact that the existence of several varieties of Guinea fowls on farms does not encourage their genetic conservation and improvement [75]. Our present findings are similar to the report of Traore et al. [76], where Guinea fowls in Burkina Faso were highly intermingled, suggesting that differences in biometric and qualitative physical traits were not related to geography. In a related study, Etienne et al. [77] reported that local Guinea fowls in Côte d'Ivoire exhibited less phenotypic diversity. In another study, it was found that Guinea fowls in northern Togo belonged to a single indigenous population [78].

## Conclusion

The qualitative physical traits of Nigerian Guinea fowls predominantly were affected by agro-ecology. However, there was no clear cut variation and distribution pattern across the three agro-ecological zones. Although the indigenous birds generally were of low body weights, those in the Southern Guinea Savanna zone were more compact while their counterparts in the Sudano-Sahelian and Tropical Rainforest zones had longer legs than body, and better condition index. Small body size could be part of the animals' adaptation for survival under the low-inputs tropical environment. The superiority of male birds to their female counterparts could be attributed to sexual dimorphism. The clustering pattern of the traits based on MCA and canonical discriminant analysis especially between the Sudano-Sahelian and Tropical Rainforest birds revealed high level of admixture, although the bird populations to an extent could be distinguished using wing length, body length, and eye colour. Overall, it could be said that the guinea fowls from the three agro-ecological zones exhibited less phenotypic diversity, and belonged to a single indigenous population. However, there is a need for further genomic studies to consolidate the present findings, and pave the way for policy decisions geared towards effective management, conservation and genetic improvement of the indigenous birds. The anticipated benefits include the development of hybrid improved Guinea fowls for the empowerment of women and youth including improvement in food security and livelihoods.

## Supporting information

**S1 Data.**
(XLS)

**S2 Data.**
(PDF)

## Acknowledgments

The authors are extremely grateful to the poultry keepers, extension agents and village heads and contact persons that facilitated data collection. We are also highly indebted to Assoc. Prof.

Şenol Çelik of the Department of Biometrics and Genetics, Faculty of Agriculture, Bingöl University, Turkey for his help with respect to the CHAID and Exhaustive CHAID analysis.

## Author Contributions

**Conceptualization:** Abdulmojeed Yakubu, Praise Jegede, Ayoola J. Shoyombo, Ayotunde O. Adebambo, Olufunmilayo A. Adebambo.

**Data curation:** Praise Jegede, Mathew Wheto, Samuel T. Vincent, Harirat L. Mundi.

**Formal analysis:** Abdulmojeed Yakubu, Praise Jegede, Samuel T. Vincent.

**Funding acquisition:** Abdulmojeed Yakubu, Mathew Wheto, Ayoola J. Shoyombo, Ayotunde O. Adebambo, Mustapha A. Popoola, Osamede H. Osaiyuwu, Olurotimi A. Olafadehan, Olayinka O. Alabi, Comfort I. Ukim, Adeniyi Olayanju, Olufunmilayo A. Adebambo.

**Investigation:** Praise Jegede, Mathew Wheto, Olayinka O. Alabi, Samuel T. Vincent, Harirat L. Mundi.

**Methodology:** Abdulmojeed Yakubu, Ayoola J. Shoyombo, Ayotunde O. Adebambo, Mustapha A. Popoola, Osamede H. Osaiyuwu, Olurotimi A. Olafadehan, Olayinka O. Alabi, Comfort I. Ukim, Adeniyi Olayanju, Olufunmilayo A. Adebambo.

**Project administration:** Abdulmojeed Yakubu, Ayoola J. Shoyombo, Ayotunde O. Adebambo, Mustapha A. Popoola, Osamede H. Osaiyuwu, Olurotimi A. Olafadehan, Olayinka O. Alabi, Comfort I. Ukim, Adeniyi Olayanju, Olufunmilayo A. Adebambo.

**Resources:** Abdulmojeed Yakubu, Mathew Wheto, Ayoola J. Shoyombo, Ayotunde O. Adebambo, Mustapha A. Popoola, Osamede H. Osaiyuwu, Olurotimi A. Olafadehan, Olayinka O. Alabi, Comfort I. Ukim, Adeniyi Olayanju, Olufunmilayo A. Adebambo.

**Software:** Abdulmojeed Yakubu, Osamede H. Osaiyuwu.

**Supervision:** Abdulmojeed Yakubu.

**Writing – original draft:** Abdulmojeed Yakubu, Praise Jegede, Mathew Wheto.

**Writing – review & editing:** Ayoola J. Shoyombo, Ayotunde O. Adebambo, Mustapha A. Popoola, Osamede H. Osaiyuwu, Olurotimi A. Olafadehan, Olayinka O. Alabi, Comfort I. Ukim, Samuel T. Vincent, Harirat L. Mundi, Adeniyi Olayanju, Olufunmilayo A. Adebambo.

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
