## [Decision Letter · Decision Letter 0]

27 Jan 2022

PONE-D-21-36760Multivariate Characterization of Morpho-biometric Traits of Indigenous Helmeted Guinea Fowl (Numida meleagris) in NigeriaPLOS ONE

Dear Dr. Yakubu,

Thank you for submitting your manuscript to PLOS ONE. After careful consideration, we feel that it has merit but does not fully meet PLOS ONE’s publication criteria as it currently stands. Therefore, we invite you to submit a revised version of the manuscript that addresses the points raised during the review process.

There are a lot of careless mistakes in the manuscript and the whole manuscript shall be reviewed by the authors carefully. Specifically the main problem found in the manuscript is related to the some aspects of methodology, statistical analysis and editing style.

We look forward to receiving your revised manuscript.

Kind regards,

Arda Yildirim, Ph.D.

Academic Editor

PLOS ONE

Journal Requirements:

2. In your Methods section, please provide additional details regarding participant consent from the owners of the animals. In the ethics statement in the Methods and online submission information, please ensure that you have specified (1) whether consent was informed and (2) what type you obtained (for instance, written or verbal). If the need for consent was waived by the ethics committee, please include this information.

“The study received financial assistance from the competitive National Research Fund of the Tertiary Education Trust Fund (TETFUND) of the Federal Republic of Nigeria through grant no TEF/DR&D/CE/NRF/UNI/ABEOKUTA/ STI/VOL.1.”

We note that you have provided funding information within the Acknowledgements Section. Please note that funding information should not appear in the Acknowledgments section or other areas of your manuscript. We will only publish funding information present in the Funding Statement section of the online submission form.

“The following authors: AY, MW, AJS, AOA, MAP, OHO, OAO, OOA, CIU,AO, OAA received funding through grant no TEF/DR&D/CE/NRF/UNI/ABEOKUTA/ STI/VOL.1. from the Tertiary Education Trust Fund (TETFUND) of the Federal Republic of Nigeria (https://tetfundserver.com/). The funders had no role in study design, data collection and analysis, decision to publish, or preparation of the manuscript.”

Additional Editor Comments:

I have finally received the experts' comments. As you can see below both found your research interesting but with some flaws that do not allow me to recommend it for publication in its current form, especially reviewer#2 comments. I suggest then to revise/amend the paper according to the reviewers' comments and resubmit for a second round of revision. Please note that in its current state, the level of English throughout your manuscript does not meet the journal’s required standard. You may wish to ask a native speaker to check your manuscript for grammar, style and syntax, or use the professional language editing options. The MS should be presented according to guidelines for authors of Plos One. Please ensure that your manuscript meets PLOS ONE's style requirements, including those for file naming. The PLOS ONE style templates can be found at

For your guidance, you can check the reviewers' comments. Thank you for giving us the opportunity to consider your work.

Reviewers' comments:

Reviewer's Responses to Questions

**Comments to the Author**

1. Is the manuscript technically sound, and do the data support the conclusions?

Reviewer #1: Partly

Reviewer #2: No

Reviewer #3: Yes

2. Has the statistical analysis been performed appropriately and rigorously? 

Reviewer #1: Yes

Reviewer #2: I Don't Know

Reviewer #3: Yes

3. Have the authors made all data underlying the findings in their manuscript fully available?

Reviewer #1: Yes

Reviewer #2: No

Reviewer #3: Yes

4. Is the manuscript presented in an intelligible fashion and written in standard English?

Reviewer #1: No

Reviewer #2: No

Reviewer #3: Yes

5. Review Comments to the Author

Reviewer #1: Quantitate analyses were done to describe and categorize the Guinea fowl characteristics.

The standard of writing should be re-considered while some of sentences are difficult to understand. This is mostly because of using some strange words like 'veritable', 'embarked'... Please use usual words and phrases instead.

What I am wondering about, after all the analysis, is how this result may help the birds producers and Poultry researchers.

I feel that the results are more of interest for Zoologist rather than Commercially poultry community.

Different ways for analyses were used. However it is not quite clear what was the point to use all the methods. The results of decision tree is not described appropriately. After each method, there may be a clear paragraph about the precision and accuracy of the models and then conclusion about basing results.

The references list should be shortened.

Reviewer #2: The reviewer read this manuscript repeatedly and carefully, nevertheless could not understand the relationship of morpho-biometric trait of the guinea fowl and "agro-ecology", which are the subjects of this manuscript.

The term "agro-ecology" which was used a lot in this manuscript, seems to be less specific and obscure the context of the entire manuscript.

Materials and methods lack a description of procedures that provide objective reproducibility. Please describe the process concretely for the materials and methods, and organize the results and considerations in the order in which the materials and methods are described.

In particular, there is no concrete description necessary to understand how to use statistical software and what analysis results lead to what conclusions.

Please unify the terms used in a unique sense.

Other points the reviewer noticed are as follows.

The reviewer could not open and view the File of Data 1-Qualitative.

Line 80: This sentence seems to indicate the purpose of the current study, but I don't understand what "qualitative traits" mean. Also, does "liner body measurement" mean measurement of specific body parts?

For example, does it mean "The current study aimed to find morphological differences in Nigerian guinea fowl based on specific quantitative indicators."?

Line 95: What does “the relative availability of indigenous” mean objectively in this study?

Also, is the "ease of data collection" appropriate as a research design for scientific verification?

Line 115: The reviewer suspected that the following points regarding the sampling may affect the results of the statistical analysis: differences in the number of bird samples extracted from each of the three defined zones, the difference of male-female ratio, the individual willingness to participate.

Please explain to get rid of these concerns.

Line 116: Please do not omit "zone".

Line 117: Is "270 birds" correct for "290 birds"?

Line 180: It can be read from the context that the zone division described by the term “the three agro-echological zone” means the three zones shown in lines 94-95. However, for more contextual clarity, the reviewer recommended defining that “the three agro-echological zones” are the three zones shown on lines 94-95.

Line 194: Please write the CHAID abbreviation first.

Reviewer #3: Comments to the Author

The manuscript entitled " Multivariate Characterization of Morpho-biometric Traits of Indigenous Helmeted Guinea Fowl (Numida meleagris) in Nigeria" represents a considerable amount of work. The following comments need to be addressed before the manuscript is suitable for publication in Plos One Journal.

Line 42: Please change Inspite of to In spite of

Line 117: There is wrong in total number of birds…..290 birds (80 males and 190 females)

Line 128: (Guinea fowl) please make the same form for Guinea fowl throughout the manuscript. Because in some parts, you write guinea fowl.

6. PLOS authors have the option to publish the peer review history of their article (what does this mean?). If published, this will include your full peer review and any attached files.

Reviewer #1: **Yes: **Hamed Ahmadi

Reviewer #2: No

Reviewer #3: No

---

## [Author Response · Author response to Decision Letter 0]

9 Mar 2022

Editorial Comments

Comment: Revisit methodology, statistical analysis and editing style 

Response: We have revisited the methodology, statistical analysis and editing style as suggested.

Comment: A rebuttal letter that responds to each point raised by the academic editor and reviewer(s). You should upload this letter as a separate file labeled 'Response to Reviewers'.

Response: A rebuttal letter has been uploaded

Comment: A marked-up copy of your manuscript that highlights changes made to the original version. You should upload this as a separate file labeled 'Revised Manuscript with Track Changes'.

Response: A copy of the Manuscript that highlights changes made has been uploaded.

Comment: An unmarked version of your revised paper without tracked changes. You should upload this as a separate file labeled 'Manuscript'.

Response: An unmarked version of the revised paper has been uploaded.

Comment: Adhere to the journal’s formatting style.

Response: We have complied with the journal’s formatting style as directed.

Comment: Provide additional details regarding participant consent from the owners of the animals. In the ethics statement in the Methods and online submission information, please ensure that you have specified (1) whether consent was informed and (2) what type you obtained (for instance, written or verbal). If the need for consent was waived by the ethics committee, please include this information.

Response: Written informed consent was obtained from the farmers. This has been included in the Ethics statement of the revised manuscript.

Comment: Remove funding information from the Acknowledgements Section

Response: We have complied as directed.

Comment: Any amendment to funding information?

Response: No, we are maintaining the existing funding information ‘‘AY, MW, AJS, AOA, MAP, OHO, OAO, OOA, CIU,AO, OAA received funding through grant no TEF/DR&D/CE/NRF/UNI/ABEOKUTA/ STI/VOL.1. from the Tertiary Education Trust Fund (TETFUND) of the Federal Republic of Nigeria (https://tetfundserver.com/). The funders had no role in study design, data collection and analysis, decision to publish, or preparation of the manuscript.”

Reviewer #1: Quantitate analyses were done to describe and categorize the Guinea fowl characteristics.

Comments: The standard of writing should be re-considered while some of sentences are difficult to understand. This is mostly because of using some strange words like 'veritable', 'embarked'... Please use usual words and phrases instead.

Response: We have revisited the whole manuscript for appropriate use of words and also subjected it to professional English editing from Senior Colleagues. 

Comments: What I am wondering about, after all the analysis, is how this result may help the birds producers and Poultry researchers. I feel that the results are more of interest for Zoologist rather than Commercially poultry community.

Response: Thanks for your observation. However, I wish to affirm that the results will be of immense usefulness to Zoologists and the Poultry Community because the first step in the genetic improvement of indigenous stock involves phenotypic characterization. It is a significant tool in breeding programs that allows the preservation of animal biodiversity and supports consumer demands. Similarly, the phenotypic traits can be further explored for improvement, as they provide useful preliminary information for genomic studies.

Comments: Different ways for analyses were used. However it is not quite clear what was the point to use all the methods. The results of decision tree is not described appropriately. After each method, there may be a clear paragraph about the precision and accuracy of the models and then conclusion about basing results.

Response: The essence of using different statistical analyses is to come up with a categorical statement with respect to the homogeneity or heterogeneity of the indigenous Guinea fowl populations in three different agro-ecological zones in Nigeria. Information that is not revealed by one statistical method may be revealed by the other, and collectively, a more meaningful and valid conclusion can be made. This will guide appropriately the process of selection of birds for genetic improvement. Based on the concern of the reviewer, we consulted a statistical expert on Decision Trees, Assoc. Prof. Şenol Çelik of the Department of Biometrics and Genetics, Faculty of Agriculture, Bingöl University, Turkey to bring out the best out of the decision trees. We have provided two additional Tables, one each for CHAID and Exhaustive CHAID to provide information on precision and accuracy including model comparison. We have also described and discussed the results appropriately as suggested. 

Comments: The references list should be shortened.

Response: We have reduced the number of references as suggested. 

Reviewer #2

Comments: The reviewer read this manuscript repeatedly and carefully, nevertheless could not understand the relationship of morpho-biometric trait of the guinea fowl and "agro-ecology", which are the subjects of this manuscript. The term "agro-ecology" which was used a lot in this manuscript, seems to be less specific and obscure the context of the entire manuscript.

Response: We have provided a map of Nigeria to show the three different agro-ecological zones for proper delineation and understanding. We have also described each agro-ecological zone based on its peculiar coordinates, climate and vegetation. This is different from the description provided in the original manuscript where we described study locations in each agro-ecological zone. Our null hypothesis is that the qualitative (phaneroptic) and quantitative (morphometric) traits of Guinea fowls in the three agro-ecological zones are statistically (P>0.05) the same. The alternative hypothesis is that the phaneroptic and morphometric traits are statistically (P<0.05) different. 

Comments: Materials and methods lack a description of procedures that provide objective reproducibility. Please describe the process concretely for the materials and methods, and organize the results and considerations in the order in which the materials and methods are described. In particular, there is no concrete description necessary to understand how to use statistical software and what analysis results lead to what conclusions. 

Response: We have provided more information on materials and methods and organized the results and discussion in the order in which the materials and methods were described. We have also expanded the description of the use of the statistical software and linked results to conclusions. The first paragraph of the revised Discussion took care of Descriptive statistics under Materials and methods (Tables 1 and 2 of Results). The second paragraph discussed Correspondence analysis (Figure 2 of Results). The third paragraph dealt with Univariate analysis (Tables 3, 4 and 5 of Results). The fourth paragraph highlighted the Stepwise canonical discriminant analysis (Figure 3, Tables 6 and 7 of Results). The fifth paragraph discussed Categorical principal component analysis (Table 8 and Figure 4 of Results) while the sixth paragraph discussed Decision trees (Figures 5 and 6, and Tables 9 and 10 of Results).

Comment: Please unify the terms used in a unique sense.

Response: We have unified the terms used in a unique sense as suggested. For instance, we have replaced qualitative traits with ‘qualitative physical traits’ throughout the text. We are also consistent with the use of ‘biometric traits and morphological indices’ to describe the quantitative traits. 

Comment: The reviewer could not open and view the File of Data 1-Qualitative.

Response: We have merged the qualitative traits and the quantitative traits data and provided a single excel file under supplementary information (S1 DATA) in the revised manuscript. We are very sorry that the initial Data 1-Qualitative file could not be opened.

Comment: Line 80: This sentence seems to indicate the purpose of the current study, but I don't understand what "qualitative traits" mean. Also, does "liner body measurement" mean measurement of specific body parts? For example, does it mean "The current study aimed to find morphological differences in Nigerian guinea fowl based on specific quantitative indicators."?

Response: By "qualitative traits", we mean those physically observable traits in animals that are discontinuous and discrete. Examples are plumage colour, skin colour, helmet shape, etc. Yes, "liner body measurement" means the same thing with biometric traits or measurements. We have recast the sentence as follows: ‘The current study aimed to find differences in indigenous Guinea fowl based on qualitative physical traits, biometric traits and morphological indices in three agro-ecological zones in Nigeria. 

Comment: Line 95: What does “the relative availability of indigenous” mean objectively in this study? Also, is the "ease of data collection" appropriate as a research design for scientific verification?

Response: Guinea fowl distribution in Nigeria is not as widespread as chicken, hence the use of ‘relative availability’. However, we have deleted the sentence together with "ease of data collection" in the course of revision. 

Comment: Line 115: The reviewer suspected that the following points regarding the sampling may affect the results of the statistical analysis: differences in the number of bird samples extracted from each of the three defined zones, the difference of male-female ratio, the individual willingness to participate. Please explain to get rid of these concerns.

Response: Based on the concern of the reviewer, we revisited the original Tables 3, 4, and 5 that reported fixed and interaction effects of the quantitative traits. Due to unequal sample sizes, low male-female ratio and non-normality of the distribution, we decided to test only fixed effect of agro-ecology, sex and sexes within each agro-ecology. For fixed effect of agro-ecology, we used the non-parametric Kruskal-Wallis H test to compare mean ranks of the biometric measurements and morphological indices. Where there were significant (P<0.05) differences, the mean ranks were separated using Dunn-Bonferroni test following the description of Brown et al. (2017). For both sex and sexes within each agro-ecology effects, we equally used non-parametric Kruskal-Wallis H test, but significant (P<0.05) mean ranks were separated using Mann–Whitney U test. We had reflected these changes under statistical analyses, presented and discussed the results appropriately in the revised manuscript. We had earlier reported in the original manuscript the use of Kruskal-Wallis H test for the qualitative physical traits. 

Comment: Line 116: Please do not omit "zone".

Response: We have included "zone". 

Comment: Line 117: Is "270 birds" correct for "290 birds"

Response: The correct figure is 270 birds. We are very sorry for the error. 

Comment: Line 180: It can be read from the context that the zone division described by the term “the three agro-echological zone” means the three zones shown in lines 94-95. However, for more contextual clarity, the reviewer recommended defining that “the three agro-echological zones” are the three zones shown on lines 94-95.

Response: For the purpose of clarity, we have provided the map of the three distinct agro-ecological zones.

Comment: Line 194: Please write the CHAID abbreviation first.

Response: We have rewritten it as ‘Chi-square automatic interaction detection (CHAID)’.

Reviewer #3

Comments to the Author The manuscript entitled " Multivariate Characterization of Morpho-biometric Traits of Indigenous Helmeted Guinea Fowl (Numida meleagris) in Nigeria" represents a considerable amount of work. The following comments need to be addressed before the manuscript is suitable for publication in Plos One Journal. 

Response: We thank the reviewer for the positive comment. We have addressed the queries as indicated below:

Comment: Line 42: Please change Inspite of to In spite of

Response: We have changed ‘Inspite of’ to ‘In spite of’.

Comment: Line 117: There is wrong in total number of birds…..290 birds (80 males and 190 females)

Response: 290 birds have been changed to 270 birds. We are very sorry for the error.

Comment: Line 128: (Guinea fowl) please make the same form for Guinea fowl throughout the manuscript. Because in some parts, you write guinea fowl.

Response: We are consistent with the use of Guinea fowl in the revised manuscript.

---

## [Decision Letter · Decision Letter 1]

28 Apr 2022

PONE-D-21-36760R1

Multivariate Characterisation of Morpho-biometric Traits of Indigenous Helmeted Guinea Fowl ( Numida meleagris ) in Nigeria

PLOS ONE

Dear Dr. Yakubu,

Thank you for submitting your manuscript to PLOS ONE. After careful consideration, we feel that it has merit but does not fully meet PLOS ONE’s publication criteria as it currently stands. Therefore, we invite you to submit a revised version of the manuscript that addresses the points raised during the review process.

It needs minor changes for the details of the purpose of the study method in the trial and the answer to the question of which bird communities are useful for further genetic reproduction. You could also access and add recent papers on guinea fowls that could contribute greatly. Thanks for sincerely and thoroughly considering and attending to the comments and concerns.

We look forward to receiving your revised manuscript.

Kind regards,

Arda Yildirim, Ph.D.

Academic Editor

PLOS ONE

Journal Requirements:

Additional Editor Comments (if provided):

Dear Authors, in your research, in which you used 74 references, you could also access and add recent papers on guinea fowls that could contribute greatly. Thanks for sincerely and thoroughly considering and attending to the comments and concerns.

Best Regards, Arda Yıldırım

Reviewers' comments:

Reviewer's Responses to Questions

**Comments to the Author**

1. If the authors have adequately addressed your comments raised in a previous round of review and you feel that this manuscript is now acceptable for publication, you may indicate that here to bypass the “Comments to the Author” section, enter your conflict of interest statement in the “Confidential to Editor” section, and submit your "Accept" recommendation.

Reviewer #1: (No Response)

Reviewer #3: All comments have been addressed

2. Is the manuscript technically sound, and do the data support the conclusions?

Reviewer #1: Partly

Reviewer #3: Yes

3. Has the statistical analysis been performed appropriately and rigorously? 

Reviewer #1: Yes

Reviewer #3: Yes

4. Have the authors made all data underlying the findings in their manuscript fully available?

Reviewer #1: No

Reviewer #3: Yes

5. Is the manuscript presented in an intelligible fashion and written in standard English?

Reviewer #1: No

Reviewer #3: Yes

6. Review Comments to the Author

Reviewer #1: My main concern mentioned in first round of review is still exist:

''Different ways for analyses were used. However it is not quite clear what was the point to use all the methods.''.

Simply, in this manuscript, there are several methods of data clustering used for handling a relatively small data set. As you know each of methods produced results for characterizing Guinea fowl. Generalization of the results are limited because we actually do not know about goal of using each methods. Sorry but I can say this is a simple data analysis using several algorithm that produced unclear results.

Answer to the question like which community of birds are useful for further genetic breeding was your priority in this study. So with your results there is no such a clear answer to that question.

Reviewer #3: All required comments have been addressed by the authors for the Manuscript Number PONE-D-21-36760R1.

7. PLOS authors have the option to publish the peer review history of their article (what does this mean?). If published, this will include your full peer review and any attached files.

Reviewer #1: **Yes: **Hamed Ahmadi

Reviewer #3: No

---

## [Author Response · Author response to Decision Letter 1]

3 May 2022

Academic Editor

General Comment: It needs minor changes for the details of the purpose of the study method in the trial and the answer to the question of which bird communities are useful for further genetic reproduction.

Response: We have appropriately responded to the reason for using different algorithms, and also answered the question as regards which bird communities are useful for further genetic reproduction.

Specific Comments

Comment: A rebuttal letter that responds to each point raised by the academic editor and reviewer(s). You should upload this letter as a separate file labeled 'Response to Reviewers'.

Response: A rebuttal letter has been uploaded.

Comment: A marked-up copy of your manuscript that highlights changes made to the original version. You should upload this as a separate file labeled 'Revised Manuscript with Track Changes'.

Response: A copy of the manuscript that highlights changes made has been appropriately labeled and uploaded.

Comment: An unmarked version of your revised paper without tracked changes. You should upload this as a separate file labeled 'Manuscript'.

Response: An unmarked version of the revised paper has been uploaded.

Comment: in your research, in which you used 74 references, you could also access and add recent papers on guinea fowls that could contribute greatly.

Response: As suggested, we have added additional four manuscripts to beef up our work.

Reviewer #1

General comments: My main concern mentioned in first round of review is still exist:

''Different ways for analyses were used. However it is not quite clear what was the point to use all the methods.''.

Simply, in this manuscript, there are several methods of data clustering used for handling a relatively small data set. As you know each of methods produced results for characterizing Guinea fowl. Generalization of the results are limited because we actually do not know about goal of using each methods. Sorry but I can say this is a simple data analysis using several algorithm that produced unclear results.

Answer to the question like which community of birds are useful for further genetic breeding was your priority in this study. So with your results there is no such a clear answer to that question.

Response: We highly appreciate the concern of the highly distinguished reviewer as regards the need to make clearer the purpose of using different algorithms, and the inference that can be drawn from the study. We used different algorithms in order to be able to arrive at a more informed decision especially the development of phenotypic standards for breeding and genetic purpose as proposed by FAO (2012). When morphometric traits are considered jointly, multifactorial analyses (using different algorithms) have been shown to assess better the within-population variation which can be utilized in the discrimination of different population types. What we have done is similar to what some of our distinguished peers have published on related topics in poultry species (Traore et al. 2018; Otecko et al., 2019; Toalombo Vargas et al., 2019; Brito et al., 2021; González Ariza, et al., 2021). However, we have included an additional paragraph where we highlighted the general implication of our findings based on the algorithms used. We reported that guinea fowls from the Southern Guinea Savanna, Sudano-Sahelian and Tropical Rainforest zones of Nigeria are more homogeneous than heterogeneous with respect to the investigated qualitative physical traits, biometric traits and morphological indices. This means that elite birds from the three agro-ecological zones are potential candidates for pure breeding or crossbreeding with their more productive exotic counterparts. Meanwhile, we sincerely welcome further suggestions from the distinguished reviewer on how to make the manuscript better.

---

## [Decision Letter · Decision Letter 2]

31 May 2022

Multivariate Characterisation of Morpho-biometric Traits of Indigenous Helmeted Guinea Fowl ( Numida meleagris ) in Nigeria

PONE-D-21-36760R2

Dear Dr. Yakubu,

We’re pleased to inform you that your manuscript has been judged scientifically suitable for publication and will be formally accepted for publication once it meets all outstanding technical requirements.

Kind regards,

Arda Yildirim, Ph.D.

Academic Editor

PLOS ONE

https://www.researchgate.net/profile/Arda-Yildirim

Additional Editor Comments (optional):

Thanks for your hard work!

Reviewers' comments:

Reviewer's Responses to Questions

**Comments to the Author**

1. If the authors have adequately addressed your comments raised in a previous round of review and you feel that this manuscript is now acceptable for publication, you may indicate that here to bypass the “Comments to the Author” section, enter your conflict of interest statement in the “Confidential to Editor” section, and submit your "Accept" recommendation.

Reviewer #1: All comments have been addressed

2. Is the manuscript technically sound, and do the data support the conclusions?

Reviewer #1: Partly

3. Has the statistical analysis been performed appropriately and rigorously? 

Reviewer #1: Yes

4. Have the authors made all data underlying the findings in their manuscript fully available?

Reviewer #1: No

5. Is the manuscript presented in an intelligible fashion and written in standard English?

Reviewer #1: Yes

6. Review Comments to the Author

Reviewer #1: Checked

no more comments.

7. PLOS authors have the option to publish the peer review history of their article (what does this mean?). If published, this will include your full peer review and any attached files.

Reviewer #1: **Yes: **Hamed Ahmadi

---

## [Editor Report · Acceptance letter]

2 Jun 2022

PONE-D-21-36760R2 

Multivariate Characterisation of Morpho-biometric Traits of Indigenous Helmeted Guinea Fowl *(Numida meleagris)* in Nigeria 

Dear Dr. Yakubu:

I'm pleased to inform you that your manuscript has been deemed suitable for publication in PLOS ONE. Congratulations! Your manuscript is now with our production department. 

Kind regards, 

on behalf of

Prof. Dr. Arda Yildirim 

Academic Editor

PLOS ONE